# Ultrastructure of the axonal periodic scaffold reveals a braid-like organization of actin rings

Stéphane Vassilopoulos [1]*, Solène Gibaud[2], Angélique Jimenez[2], Ghislaine Caillol[2] & Christophe Leterrier [2]*

Recent super-resolution microscopy studies have unveiled a periodic scaffold of actin rings regularly spaced by spectrins under the plasma membrane of axons. However, ultrastructural details are unknown, limiting a molecular and mechanistic understanding of these enigmatic structures. Here, we combine platinum-replica electron and optical super-resolution microscopy to investigate the cortical cytoskeleton of axons at the ultrastructural level. Immunogold labeling and correlative super-resolution/electron microscopy allow us to unambiguously resolve actin rings as braids made of two long, intertwined actin filaments connected by a dense mesh of aligned spectrins. This molecular arrangement contrasts with the currently assumed model of actin rings made of short, capped actin filaments. Along the proximal axon, we resolved the presence of phospho-myosin light chain and the scaffold connection with microtubules via ankyrin G. We propose that braided rings explain the observed stability of the actin-spectrin scaffold and ultimately participate in preserving the axon integrity.

[1] Sorbonne Université, INSERM, Institute of Myology, Centre of Research in Myology, UMRS 974, Paris, France. [2] Aix Marseille Université, CNRS, INP UMR7051, NeuroCyto, Marseille, France. *email: s.vassilopoulos@institut-myologie.org; christophe.leterrier@univ-amu.fr

N euorons develop an intricate axonal arborization to ensure the proper flow and processing of information. This extraordinary architecture is built and maintained by a unique organization of the axonal cytoskeleton[1,2]. Pioneering electron microscopy (EM) studies have revealed parallel arrays of microtubules within axons that support vesicular transport over long distances[3]. By contrast, the organization of actin inside the axon shaft has long been overlooked[4]. Only recently has optical super-resolution microscopy been able to unveil striking actin assemblies within axons:[5] the axon is lined by a membrane-associated periodic scaffold (MPS) composed of circumferential actin rings spaced every ~ 185 nm by tetramers of spectrins[6–9]. A molecular and mechanistic understanding of actin rings and the MPS is still lacking, as EM has not yet visualized this assembly despite its presence in virtually all neurons[10–13]. The MPS has been proposed to provide mechanical resistance to axons[14] and to recruit components for cell signaling[15], but further exploring and understanding the MPS functions require elucidating its molecular organization.

We thus set out to investigate the ultrastructure of the axonal actin-spectrin scaffold, performing EM observation of the MPS and combining it with optical super-resolution microscopy to reveal its molecular architecture. From platinum-replica EM images of the MPS, we find that actin rings are made of long, braided actin filaments connected by a mesh of spectrins. We visualize the association of specific components with the MPS along the axon initial segment (AIS): phospho-myosin light chain is apposed to actin rings, whereas ankyrin G links the MPS to cortical microtubules via its carboxyterminal tail. The MPS organization is robust, but can be impacted by strong actin perturbations. Finally, we directly visualize the actin rings and MPS components by correlative super-resolution microscopy and platinum-replica EM, unambiguously revealing their molecular organization along the axonal plasma membrane.

## Results

**Mechanical unroofing of cultured neurons preserves the MPS.** To obtain three-dimensional views of the axonal cortical cytoskeleton in a close to native state, without the need for detergent extraction or sectioning[16,17], we mechanically unroofed cultured hippocampal neurons using ultrasound[18]. We first verified that the periodic organization of the MPS was still detected along the axon of unroofed neurons by super-resolution fluorescence microscopy based on single-molecule localization (SMLM). Staining of unroofed cultures without permeabilization highlighted the unroofed neurons with accessible ventral membrane (Fig. 1a). In unroofed proximal axons, we detected ~ 180 nm periodic rings for actin (Fig. 1b). A similar ~ 190 nm periodic pattern was also obtained when labeling the carboxyterminus of β4-spectrin along the AIS (Fig. 1c) and β2-spectrin along the more distal axon, although it was more difficult to unroof distal processes (Fig. 1d). A strong periodic pattern resulted from combining antibodies against axonal spectrins (α2, β2, and β4)[19] (Fig. 1e). We measured no difference with non-unroofed, permeabilized neurons (Supplementary Fig. 1a–g) after quantification of the MPS spacing and periodicity using autocorrelation measurements from SMLM images (Fig. 1f–h), except a more-marked periodicity of actin along the AIS as unroofing removed the intra-axonal actin patches[20].

**Platinum-replica EM reveals actin rings as braided filaments.** We then imaged unroofed neurons by transmission EM of metal replicas[21]. To identify the axonal process stemming from unroofed cell bodies, we located the characteristic fascicles of AIS microtubules (yellow, Fig. 1i)[22]. High-magnification views of the

axonal membrane-associated cytoskeleton revealed the presence of long, parallel actin filaments, oriented perpendicular to the axis of the axon and to microtubules, and often regularly spaced by ~ 200 nm. These structures resembled braids made of two actin filaments (magenta, Fig. 1j–k, Supplementary Fig. 1h, Supplementary Movies 1–3). Their average spacing of 184 ± 4 nm (mean ± SEM, Fig. 1l), and their detection only along the axon (Fig. 1m) strongly suggest that these braids are the periodic actin rings observed by SMLM.

To show that these braids were indeed made of actin filaments, we labeled axonal actin for platinum-replica EM (PREM) using two complementary methods. First, we used phalloidin-Alexa Fluor 488 staining followed by immunogold labeling[16] and observed numerous gold beads decorating the actin braids (Fig. 2a). Second, we used myosin subfragment 1 (S1) to decorate actin into a ropelike double helix[21]. Myosin S1 extensively decorated the actin braids (Fig. 2b) and was able to protect them during fixation and processing, as their average observable length went from 0.68 ± 0.04 μm to 1.13 ± 0.04 μm (Fig. 2c). High-magnification views of individual actin braids confirmed that braids are most often composed of two long (>0.5 μm) and intertwined actin filaments (Fig. 2d, e). The filaments sometimes separate at the visible extremity of a braid, bifurcating in a Y shape (Fig. 2d, e). This could result from the unwinding of a ring broken into braids during the unroofing procedure, as it is rarer along the myosin S1-protected braids. Alternatively, it may be a snapshot of actin filaments assembling into rings, as ring branching has been described along the proximal axon of living neurons[8]. We then confirmed that braids are made of two actin filaments by measuring the diameter of individual actin braids before and after splitting (Fig. 2f): braids are 18.2 ± 0.3 nm thick. When splitting, they form two 10.2 ± 0.3 nm filaments, similar in diameter to single-actin filaments in dendrites (9.9 ± 0.2 nm), and to the reported value of 9–11 nm for a single-actin filament rotary-replicated with ~2 nm of platinum[21,23]. The organization of rings as braids of long, unbranched filaments is to our knowledge unique among cellular actin structures[24] and contrasts with the currently assumed view that axonal actin rings made of short, capped and bundled actin filaments[6,25].

**Actin braids are connected by an organized mesh of spectrins.** Between actin rings, the periodic organization of spectrins implies that 190-nm long spectrin tetramers connect two adjacent rings[6,26] (Supplementary Fig. 1f–g). On our PREM images, actin braids are connected by a dense mesh of rods aligned perpendicular to the braids (yellow, Fig. 2g, Supplementary Movie 4). To identify the components of this mesh, we used immunogold labeling against β4-spectrin. Gold beads decorated the mesh filaments in-between actin braids (Fig. 2h). The immunolabeling efficiency along the unroofed proximal axons was low for β2-spectrin, as it is mostly present along the distal axon (Supplementary Fig. 2a). Similarly, we could not reliably detect adducin, as it is mostly absent from the AIS and proximal axon, as previously reported[16]. To better detect spectrins and confirm that they form the connecting mesh, we used a triple immunogold labeling of α2-, β2-, and β4-spectrin. This resulted in a number of gold beads decorating the submembrane mesh, often half-way between the actin braids (Fig. 2i). Overall, the labeling experiments demonstrate that PREM resolves axonal actin rings made from pairs of long, intertwined filaments, and reveal the dense alignment of spectrins that connect these rings to form the axonal MPS.

**Ultrastructural organization of proximal MPS components.** We next took advantage of PREM of unroofed neurons to

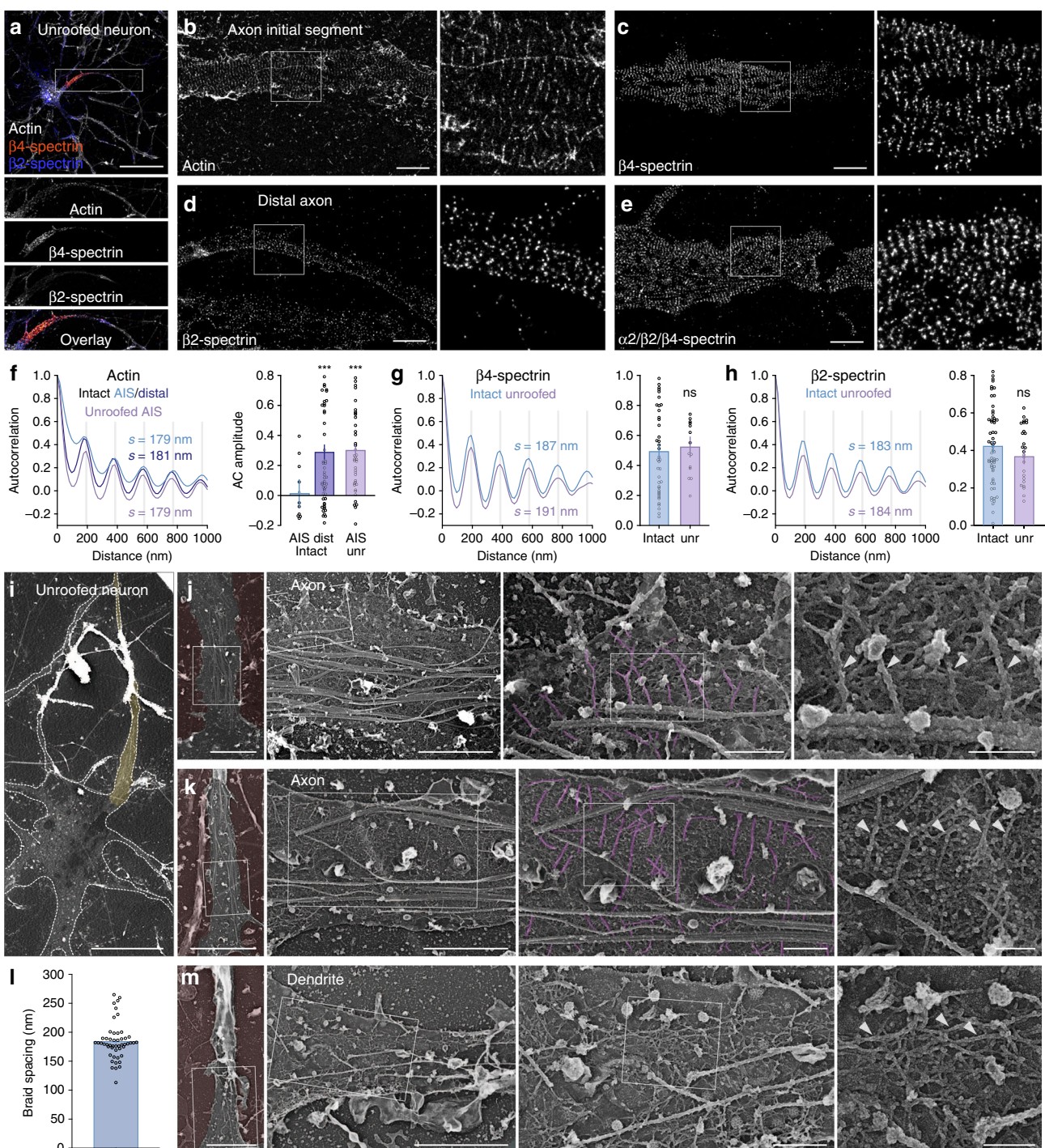

**Fig. 1 The actin-spectrin MPS is conserved in unroofed axons and visible by PREM. a** Epifluorescence image of an unroofed neuron labeled for actin (gray), β4-spectrin (orange), and β2-spectrin (blue). **b–e** SMLM images showing the periodic pattern of actin **b**, β4-spectrin **c**, β2-spectrin **d**, and α2/β2/ β4-spectrin **e** along unroofed axons. **f–h** Left, autocorrelation curve of the labeling for actin **f**, β4-spectrin **g**, or β2-spectrin **h**. Spacing (s) is indicated. Right, measurement of the corresponding autocorrelation amplitude (mean ± SEM, $n = 18$–42 tracings from 4–6 independent experiments, ns non-significant, ***$p < 0.001$, ANOVA post-hoc test). **i** Low-magnification PREM view of an unroofed neuron and its axon (yellow). **j–k** PREM views of an unroofed axon showing the regularly spaced braids (magenta, arrowheads) perpendicular to microtubule fascicles. **l** Distance between regularly spaced actin braids in axons measured on PREM views (mean ± SEM, $n = 50$ braids from five independent experiments). **m** PREM view of an unroofed dendrite from the same neuron shown in **k** containing mostly longitudinal actin filaments (arrowheads). Scale bars 40 μm **a**, 2 μm **b–e**, 20 μm **i**, 5, 2, 0.5, and 0.2 μm **j–k**, **m**, left to right). Source data for graphs **f–h**, **l** are provided as a Source Data file.

determine the ultrastructural localization of MPS components known to specifically localize along the AIS[22]. The phosphorylated myosin light chain (pMLC), an activator of myosin II, was recently reported to associate with the MPS along the initial

segment[27], together with its partner tropomyosin[28]. We found concentrated pMLC at the AIS, with areas showing a ~200 nm periodic pattern on SMLM images (Fig. 3a, b). After unroofing, a fraction of the pMLC staining was retained, although a periodic

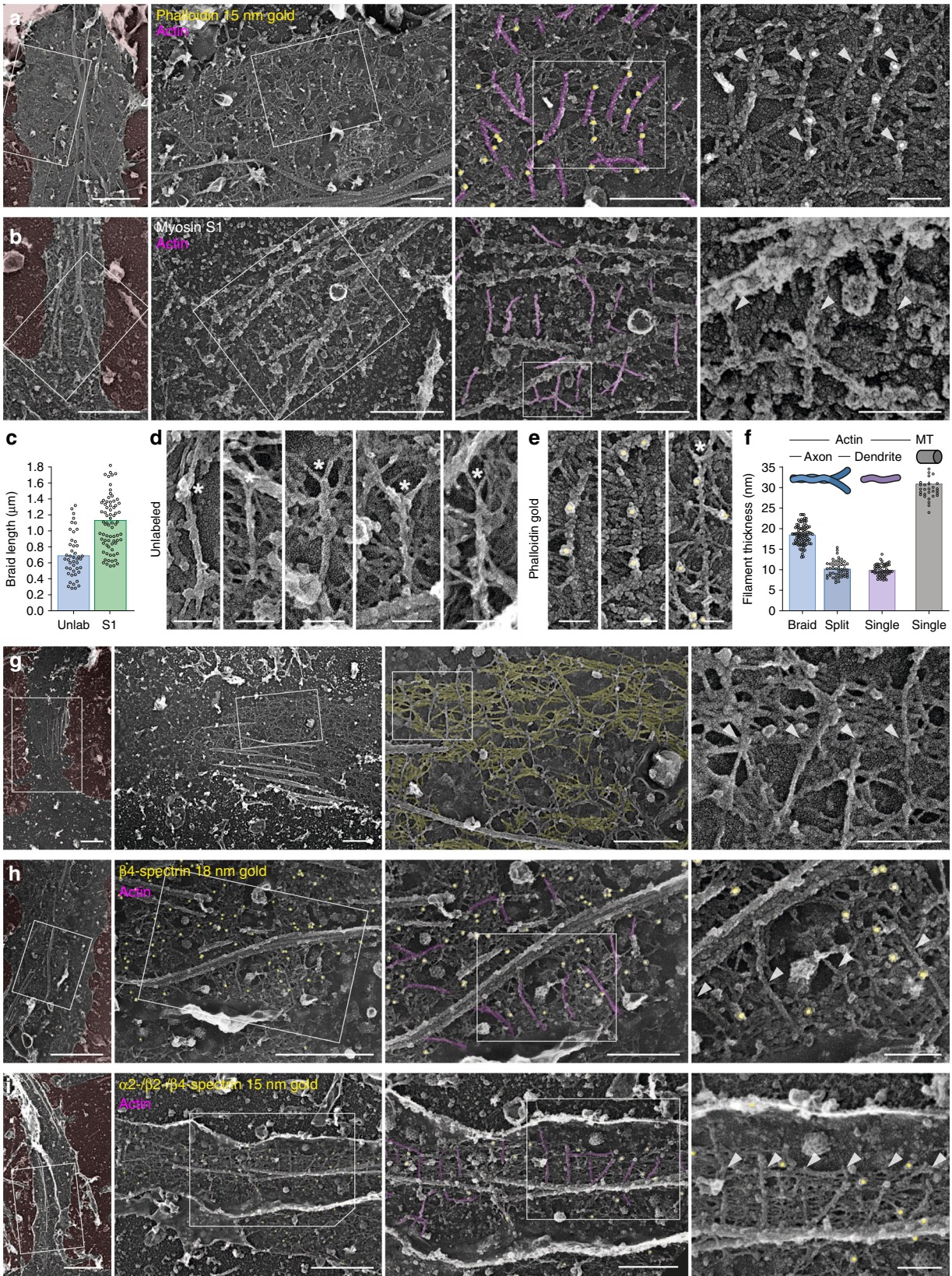

pattern was not easily detectable (Fig. 3c, d). Immunogold labeling for pMLC followed by PREM showed gold beads often appearing along filaments perpendicular to actin braids (Fig. 3e, brackets). These filaments could be myosins associated with the MPS, and suggest that myosins can cross-link neighboring rings[29].

Next, we examined the localization of ankyrin G (ankG), a large scaffold protein that recruits other components during initial segment assembly[22,30], and is required for the presence of microtubule fascicles along the AIS[31,32]. Immunolabeling in non-permeabilized conditions after unroofing could not detect the aminoterminal part of ankG or associated AIS membrane proteins

**Fig. 2 Actin rings are braids of long filaments connected by a spectrin mesh. a** PREM views of axonal actin braids (magenta, arrowheads) labeled with fluorescent phalloidin and immunogold detection of the fluorophore (15 nm gold beads, pseudo-colored yellow). **b** PREM views of myosin S1-treated axonal actin braids (magenta, arrowheads). **c** Length of the actin braids measured on PREM views in unlabeled (unlab) and myosin S1-labeled (S1) axons (mean ± SEM, $n = 45$–76 braids from 2 to 5 independent experiments). **d, e** High-magnification views of individual unlabeled **d** and immunogold-labeled **e** actin braids showing Y bifurcations (asterisks). **f** Thickness of filaments measured on PREM views: axonal actin braids before (braid) and after (split) splitting (blue), dendritic single-actin filaments (purple) and axonal microtubules (gray) (mean ± SEM, $n = 33$–90 braids from 3 to 8 independent experiments). **g** PREM views of unroofed axons showing the mesh (yellow) connecting actin braids. **h** PREM views of unroofed axons immunogold-labeled (yellow) for β4-spectrin between actin braids (magenta, arrowheads). **i** PREM views of unroofed axons simultaneously immunogold-labeled (yellow) for α2/β2/β4-spectrin between actin braids (magenta, arrowheads). Scale bars 2 μm, 1, 0.5, 0.2 μm **a**–**b**, **g**–**i**, left to right), 0.1 μm **d**, **e**. Source data for graphs **c** and **f** are provided as a Source Data file.

such as Nav channels. We thus focused on the carboxyterminal part of ankG that extends below the MPS[9] and interacts with microtubule-associated End-Binding proteins (EBs)[33] via SxIP motifs in its tail domain[34]. Labeling of the 480 kDa ankG tail[32] resulted in dense staining along the AIS in intact and unroofed neurons (Fig. 3f–i), often showing a pattern of longitudinal tracks along unroofed axons (Fig. 3i, brackets). Immunogold labeling and PREM showed a high concentration along the AIS, with beads densely decorating the spectrin mesh (Fig. 3j). AnkG labeling could be detected along the connecting spectrins in-between actin braids (Fig. 3j), consistently with the current model of AIS architecture[9]. Overall, these experiments show how PREM on unroofed neurons provides access to the AIS molecular architecture and can visualize identified proteins within its membrane-bound scaffold.

**Impact of actin perturbation on the MPS ultrastructure**. We next assessed the effect of drugs that potently target actin assemblies on the MPS organization down to the ultrastructural level. Owing to the reported stability of the actin rings[9,35], we used swinholide A, a drug that inhibits actin polymerization and severs existing filaments[36]. We also used cucurbitacin E, a drug that inhibits filament depolymerization, stabilizes them, and is compatible with phalloidin staining[37]. At the diffraction-limited level, acute swinholide treatment resulted in a near disappearance, whereas cucurbitacin induced a marked increase, of the actin staining throughout neuronal processes, with a moderate effect on spectrins labeling (Supplementary Fig. 3a–b). Using SMLM, we found that swinholide could only partially disorganize the MPS along the AIS, with a periodic appearance of the remaining actin and of β4-spectrin (Fig. 4a, b, Supplementary Fig. 3c–d), whereas in the distal axon both actin and β2-spectrin lost their periodicity (Fig. 4c, d, Supplementary Fig. 3e–f). This higher stability of the MPS in the initial segment compared with the distal axon is consistent with previous results[9,35] and may be caused by the high density of ankG/membrane protein complexes anchored within the initial segment MPS (Fig. 3j). Stabilization by cucurbitacin resulted in the presence of numerous bright longitudinal actin bundles and clusters along the axon, making quantification of the eventual ring stabilization difficult (Fig. 4a, c, Supplementary Fig. 3c, e). Cucurbitacin did not change the β4-spectrin pattern, but β2-spectrin showed more-consistent bands and a larger autocorrelation amplitude of the pattern[38] (Fig. 4b, c, Supplementary Fig. 3d, f). PREM views of treated, unroofed neurons (Fig. 4e) showed that swinholide induced a disappearance of the transverse actin braids, whereas the spectrin mesh remained visible along the proximal axon but was partially disorganized, consistent with the SMLM results (Fig. 4f). The reinforcement of the MPS induced by cucurbitacin was visible by PREM, with the occurrence of very regular stretches of actin braids connected by a densified mesh of spectrins (Fig. 4g). In addition to the similar effect detected by SMLM and PREM, these experiments show that the stable actin braids/rings can be

modulated by acute perturbations, suggesting a regulated assembly and turnover.

**Resolving actin rings by correlative SMLM/PREM**. The braids seen by PREM are not as long and continuous along the axon as the actin rings visible in SMLM (compare Fig. 1b, j, k). To explain this, we performed correlative SMLM and PREM on unroofed neurons. After sonication, fixation, and labeling for actin and spectrin, proximal axons were imaged by SMLM before being replicated, relocated on the EM grid and observed[39]. The nanoscale precision provided by both techniques allows to closely register the SMLM images and PREM views. Phalloidin staining of actin rings on SMLM images (Fig. 5a, d) frequently corresponded with the braids visible on PREM views (Fig. 5b, e). A few actin filaments on PREM images were only faintly stained by phalloidin, likely owing to the fast labeling protocol used for our correlative approach. SMLM often showed single rings across the whole width of the axon, whereas they only appeared intermittently on the PREM views (Fig. 5c, f, Supplementary Movie 5). This may result from ultrastructural damage during SMLM imaging or sample processing for subsequent PREM. However, close examination of PREM views—that only visualize the surface of the replicated sample—suggests that phalloidin-stained actin rings are often embedded between the plasma membrane and the spectrin mesh, and thus partially hidden from PREM view (Fig. 5c, f). Phalloidin staining also revealed that rings were continuous under microtubules, although they are similarly hidden from PREM view (Fig. 5c, f).

**Identification of MPS components by correlative SMLM/ PREM**. We applied the same strategy to localize spectrins and other MPS components along unroofed axons. SMLM of unroofed neurons labeled for β4-spectrin using an antibody labeling the center of the spectrin tetramer[9] exhibited well-defined bands along the AIS (Fig. 6a, Supplementary Fig. 4a, d). These bands corresponded to the spectrin mesh connecting actin braids on PREM views (Fig. 6b, Supplementary Fig. 4b, e), and often localized in-between actin braids (Fig. 6c, Supplementary Fig. 4b, e, Supplementary Movie 6). When imaging β2-spectrin, the labeling along the proximal axon was fainter by SMLM (Supplementary Fig. 5a), consistent with the presence of β2-spectrin along the more distal axon[19]. This labeling localized to the spectrin mesh in the corresponding PREM views (Supplementary Fig. 5b, c, Supplementary Movie 7). We also located pMLC by correlative SMLM/PREM and could resolve spots of pMLC labeling by SMLM (Supplementary Fig. 6a). Interestingly, these spots were often present as pairs, associated with thicker filaments roughly perpendicular to actin braids on PREM views (Supplementary Fig. 6b, c, brackets), in line with the putative myosin labeling obtained by immunogold labeling (Fig. 3e, brackets).

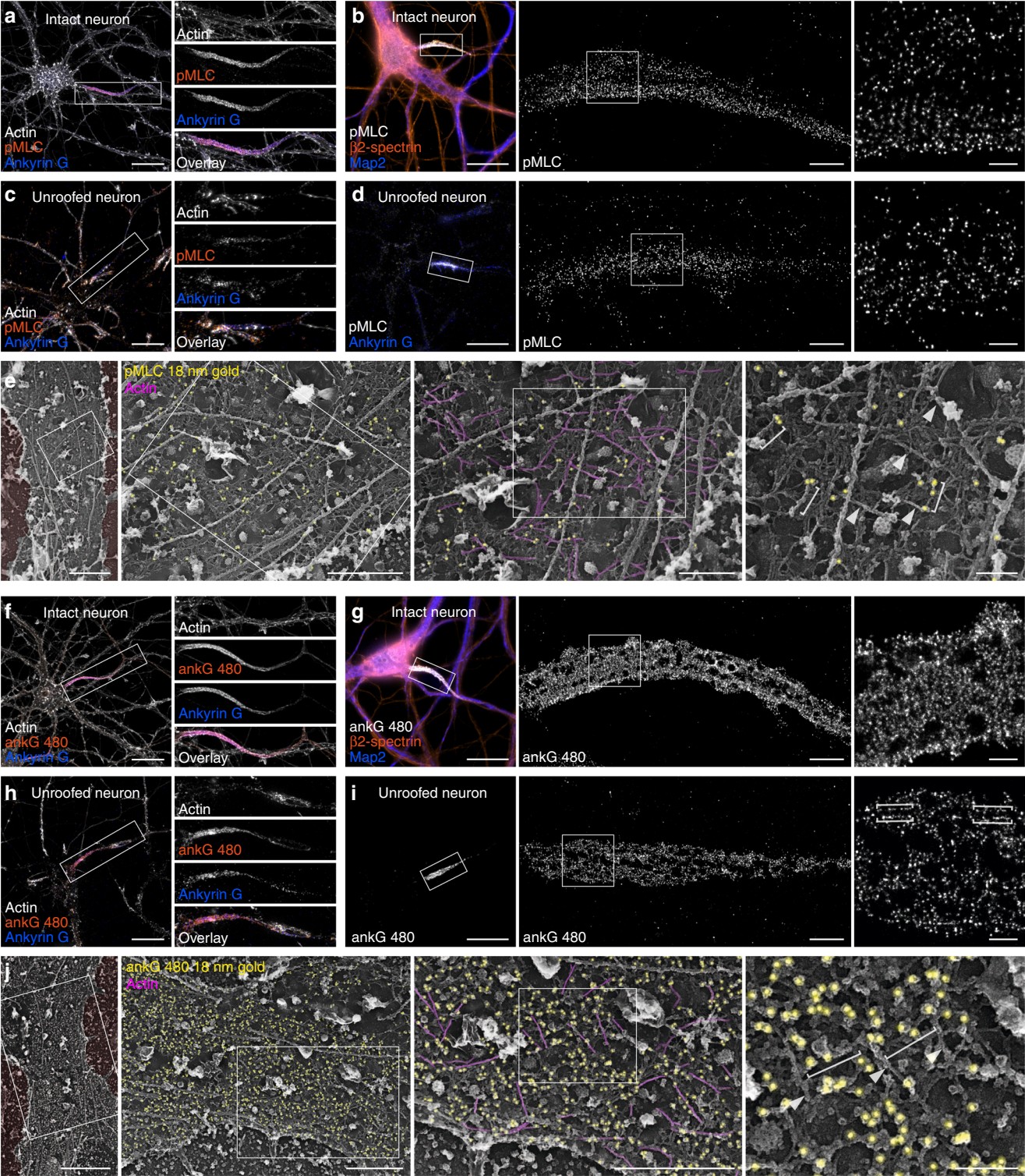

**Fig. 3 Localization of pMLC and ankyrin G at the AIS of unroofed neurons. a** Epifluorescence image of an intact neuron labeled for actin (gray), pMLC (orange), and ankyrin G (blue). **b** SMLM images showing the pattern of pMLC along an intact AIS. **c** Epifluorescence image of an unroofed neuron labeled for actin (gray), pMLC (orange), and ankyrin G (blue). **d** SMLM images showing the pattern of pMLC along an unroofed AIS. **e** PREM views of an unroofed AIS immunogold-labeled (yellow) for pMLC (brackets) apposed to actin braids (magenta, arrowheads). **f** Epifluorescence image of an intact neuron labeled for actin (gray), 480 kDa ankyrin G tail (ankG 480, orange), and ankyrin G (blue). **g** SMLM images showing the pattern of ankG 480 along an intact AIS. **h** Epifluorescence image of an unroofed neuron labeled for actin (gray), ankG 480 (orange), and ankyrin G (blue). **i** SMLM images showing the pattern of ankG 480 along an unroofed AIS, delineating the profiles of putative microtubules (brackets). **j** PREM views of an unroofed AIS immunogold-labeled (yellow) for ankG 480 appearing along spectrin filaments (brackets) in between actin braids (magenta, arrowheads). Scale bars 20 μm **a**, **c**, **f**, **h**; 2 μm, 0.5 μm **b**, **d**, **g**, **i**, left to right; 2 μm, 1, 0.5, 0.2 μm **e**, **j**, left to right.

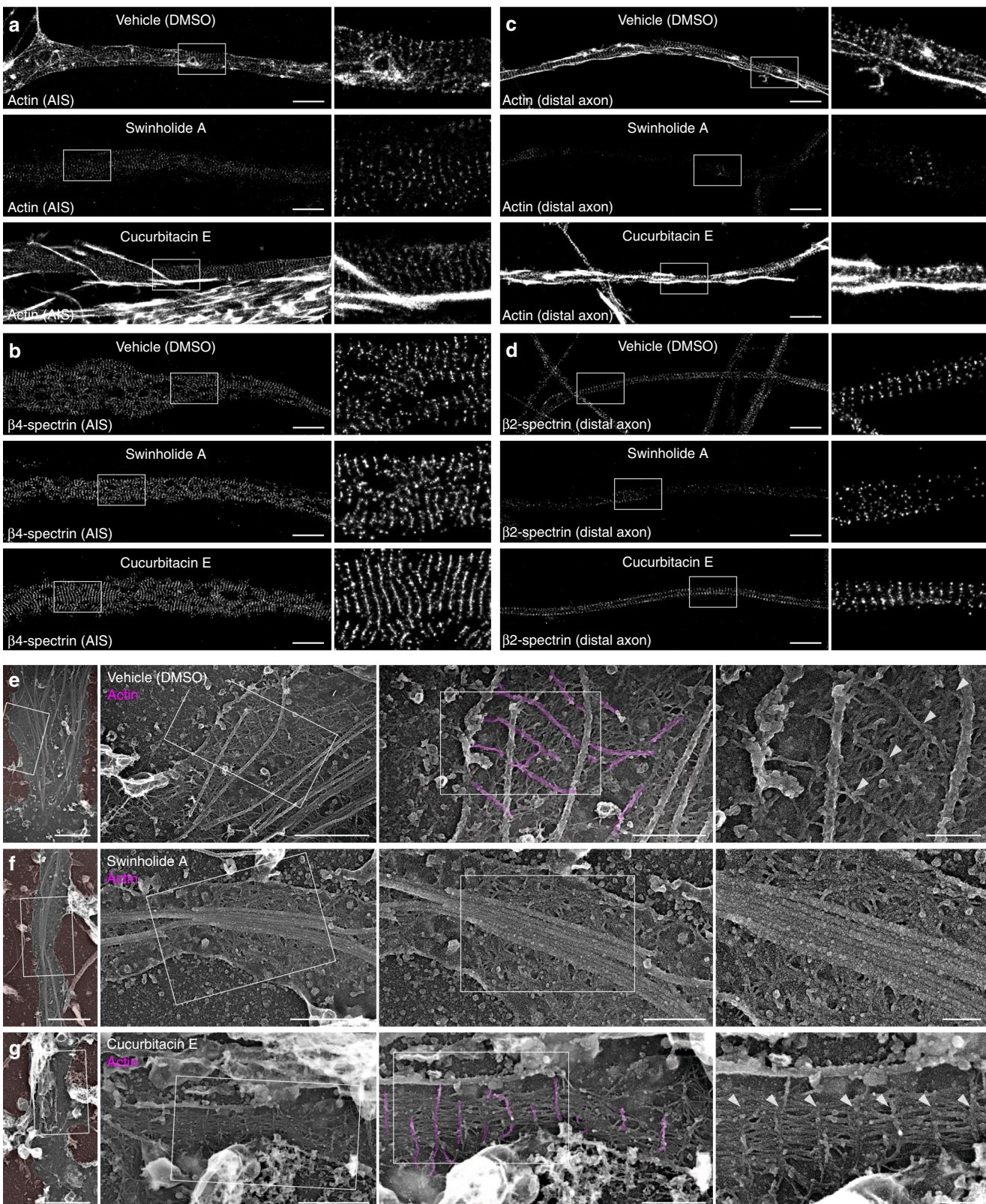

**Fig. 4 Actin perturbation impacts the MPS ultrastructure. a–d** SMLM images of AIS and distal axons treated for 3 h with vehicle (DMSO 0.1%), swinholide A (100 nM) or cucurbitacin E (5 nM), labeled for actin **a** and **c**, β4-spectrin **b**, and β2-spectrin **d**. **e–g** PREM views of unroofed axons from neurons treated with vehicle **e**, swinholide A **f**, or cucurbitacine E **g** showing the presence or absence of actin braids (magenta, arrowheads). Scale bars 2 μm **a–d**, 2 μm, 1, 0.5, 0.2 μm **e–g**, left to right.

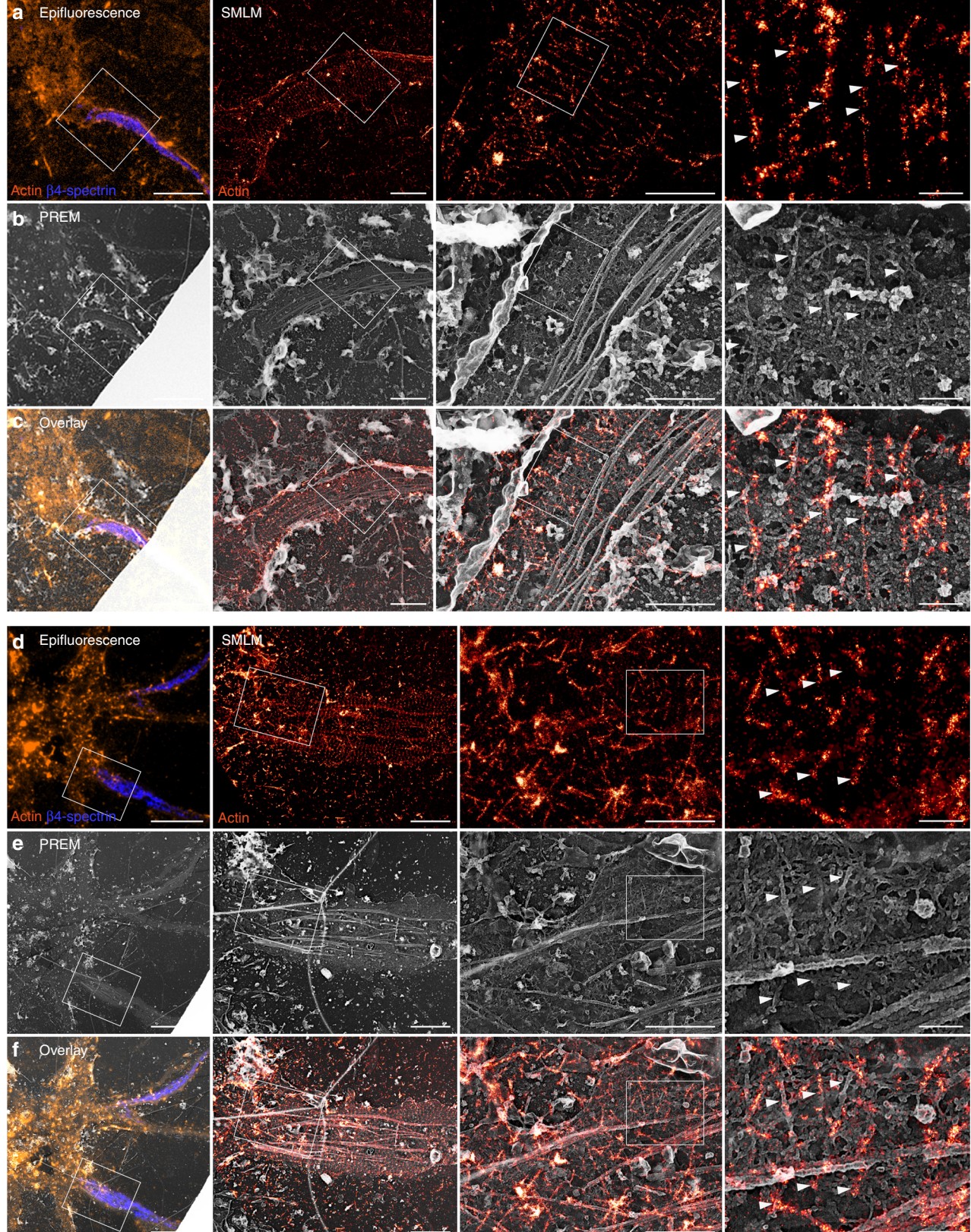

**Fig. 5 Correlative SMLM/PREM resolves the ultrastructure of actin rings. a** Left, epifluorescence image of an unroofed neuron labeled for actin (orange) and β4-spectrin (blue). Right, SMLM images of the unroofed proximal axon labeled for actin. **b** Corresponding PREM views of the same unroofed neuron and axon. **c** Overlay of the SMLM image and PREM views showing the correspondence between actin rings in SMLM and braids in PREM (arrowheads). **d–f** Correlative SMLM/PREM images similar to **a–c** for an additional unroofed axon. Scale bars 20, 2, 1, 0.2 μm (from left to right).

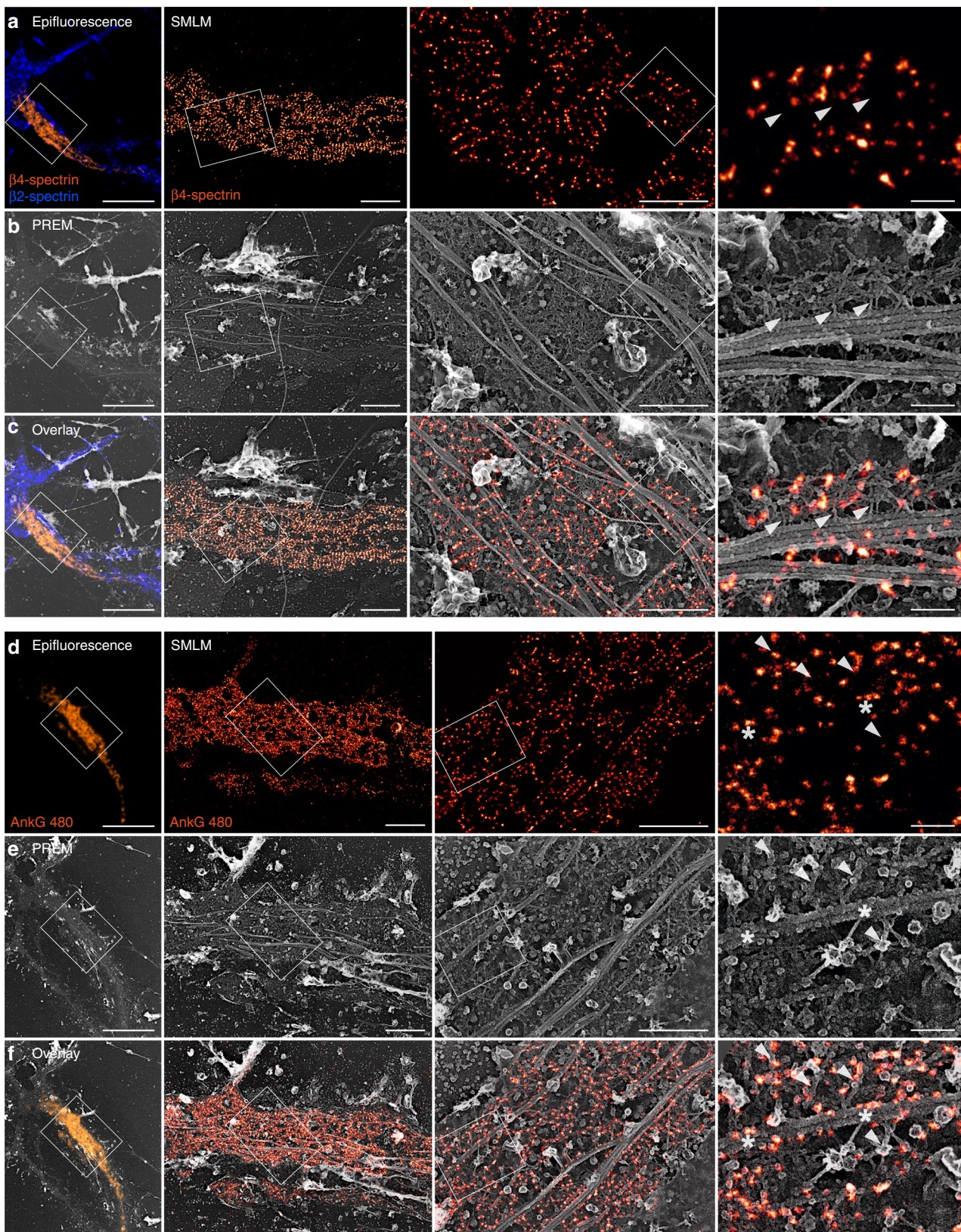

**Fig. 6 Correlative SMLM/PREM of β4-spectrin and ankyrin G.** Left, epifluorescence image of an unroofed neuron labeled for β4-spectrin (orange) and β2-spectrin (blue). Right, SMLM images of the unroofed proximal axon labeled for β4-spectrin. **b** Corresponding PREM views of the same unroofed neuron and axon. **c** Overlay of the SMLM image and PREM views showing the intercalation between spectrin tetramer centers in SMLM and actin breads in PREM (arrowheads). **d–f** Correlative imaging of 480 kDa ankyrin G tail (ankG 480) along an unroofed proximal axon by SMLM and PREM. Actin braids are indicated by arrowheads, and contacts between ankyrin G and microtubules are shown by asterisks. Scale bars 20, 2, 1, 0.2 μm (from left to right).

Finally, we correlated labeling for the 480 kDa ankG tail with PREM views of the same proximal axon (Fig. 6d–f). The density of the labeling resulted in a mesh largely covered by primary and secondary IgGs (Fig. 6e). Correlative overlay revealed that the longitudinal tracks (empty lines flanked by a high density of ankG labeling, Fig. 3i) that we observed on SMLM images correspond to microtubules decorated on both sides with fluorescent molecules that likely correspond to ankG tails (Fig. 6f). We thus directly visualize the MPS association with microtubules via ankG along the AIS, initially proposed from the interaction of ankG with microtubules End-Binding proteins EB1 and EB3 (ref. [33]). This connection could also explain why we usually detected well-preserved actin braids along unroofed areas where microtubule bundles are still present (Fig. 1j, k).

## Discussion

In conclusion, we obtained EM images of the actin-spectrin MPS along axons and provide ultrastructural insight into its molecular organization. Previous work using PREM to visualize the AIS did not detect regularly spaced actin filaments or other ultrastructural signs of periodic organization[16]. However, this work used live extraction by a detergent before fixation, an approach that was later shown to disrupt actin rings and the MPS[7]. The fact that mechanical unroofing preserves the submembrane assemblies better than detergent extraction suggests that their interaction with the membrane is essential for their stability.

We show that axonal actin rings are made of a small number of long, intertwined filaments rather than by a large number of short, bundled filaments as previously hypothesized[6,12,40,41]. This short filament model was inferred from the presence of adducin at actin rings, at least along the distal axon[6,25], because adducin was described as a capping protein that can bind to the barbed end of filaments[42]. However, adducin is also able to associate to the side of filaments, enhancing the lateral binding of spectrin to actin[43]. Although we were not able to directly visualize adducin at the ultrastructural level, our results suggest that this lateral binding role is dominant to enhance actin rings interaction with spectrins. Further studies are needed to clarify the precise localization and role of adducin as well as to confirm the localization of myosin filaments associated with the MPS, in order to explain how actin-associated proteins can regulate radial and longitudinal tension along the axon as well as axonal diameter[25,29].

Finally, the arrangement of actin rings as a braid of long filaments is relevant to the observed stability of the MPS: closely apposed, intertwined filaments are likely to be resistant to severing factors such as ADF/cofilin[44], and further stabilized by their embedding in the spectrin mesh. Rings made of long intertwined filaments are likely to be stiffer than bundled short filaments, hence could form the rigid part of a flexible and resistant scaffold when linked by elastic spectrins[41], explaining the role of the MPS in the mechanical resistance of axons[14]. Beyond shedding light on the molecular underlaying of the MPS function, the ultrastructural insights obtained here could help understand its potential dysfunctions in neurodegenerative diseases, where the axonal integrity is often affected first[45,46].

## Methods

**Antibodies and reagents**. Rabbit polyclonal anti βIV-spectrin antibody (against residues 2237–2256 of human βIV-spectrin, 1:800 dilution for immunofluorescence IF, 1:20 for immunogold IG) was a gift from Matthew Rasband (Baylor College of Medicine, Austin, TX). Mouse monoclonal anti βII-spectrin (against residues 2101–2189 of human βII-spectrin, 1:100 for IF, 1:20 for IG) was from BD Biosciences (#612563)[6]. Chicken anti-map2 antibody was from abcam (#ab5392, 1:1000 for IF). Mouse monoclonal antibodies anti ankyrin G (clone 106/65 and 106/36, 1:300 for IF) were from NeuroMab[9]. Rabbit polyclonal anti 480-kDa ankG (residues 2735–2935 of rat 480-kDa ankG, 1:300 for IF, 1:100 for IG) was a gift from Vann Bennett (Duke University, Durham, NC)[32]. Rabbit anti phospho-Myosin

Light Chain 2 Thr18/Ser19 (pMLC) was from Cell Signaling Technologies (#3674, 1:50 for IF, 1:20 for IG)[27].

Rabbit polyclonal anti-Alexa Fluor 488 antibody was from Thermo Fisher (A11094, 1:20 for IG). Donkey and goat anti-rabbit and anti-mouse secondary antibodies conjugated to Alexa Fluor 488, 555, and 647 were from Life Technologies or Jackson ImmunoResearch (1:200–1:400 for IF). Donkey anti-rabbit and anti-mouse secondary antibodies conjugated to DNA-PAINT handles P1 (ATACATCTA) and P3 (TCTTCATTA), respectively, were prepared according to published procedures[47]. Goat anti-rabbit and anti-mouse secondary antibodies conjugated to 15 nm gold nanobeads were from Aurion (#815011 and #815022, respectively, 1:20 for IG) and the ones conjugated to 18 nm gold nanobeads were from Jackson ImmunoResearch (#111-215-144 and #115-215-146, respectively, 1:15 for IG).

Alexa Fluor 488 and Alexa Fluor 647 conjugated phalloidins were from Thermo Fisher (#A12379 and #A2287, respectively), Atto488-conjugated phalloidin (#AD488-81) was from Atto-Tec. DMSO (#D2650), swinholide A (#S9810), cucurbitacine E (#PHL800-13), glutaraldehyde (#G5882) were from Sigma. Paraformaldehyde (PFA, #15714, 32% in water) was from Electron Microscopy Sciences. Myosin S1 (from rabbit skeletal fast muscle) was from Hypermol (#9310-01).

**Animals and neuronal cultures**. The use of Wistar rats followed the guidelines established by the European Animal Care and Use Committee (86/609/CEE) and was approved by the local ethics committee (agreement D13-055-8). Rat hippocampal neurons were cultured following the Banker method, above a feeder glia layer[48]. Rapidly, 12 or 18 mm-diameter round, #1.5 H coverslips were affixed with paraffine dots as spacers, then treated with poly-L-lysine. Hippocampi from E18 rat pups were dissected, and homogenized by trypsin treatment followed by mechanical trituration and seeded on the coverslips at a density of 4000–8000 cells/cm² for 3 h in serum-containing plating medium. Coverslips were then transferred, cells down, to petri dishes containing confluent glia cultures conditioned in B27-supplemented Neuro-basal medium and cultured in these dishes for up to 4 weeks. For this work, neurons were fixed at 13–17 days in vitro, a stage where they exhibit a MPS along virtually all axons[6].

**Cell treatments and fluorescence immunocytochemistry**. Treatments were applied on neurons in their original culture medium for 3 h at 37 °C, 5% CO₂: dimethyl sulfoxide (DMSO) 0.1% (from pure DMSO), swinholide A 100 nM (from 100 μM stock in DMSO), cucurbitacin E 5 nM (from 5 μM stock in DMSO). Stock solutions were stored at −20 °C.

Fluorescent immunocytochemistry for epifluorescence microscopy and SMLM was performed as in published protocols[49]. Cells were fixed using 4% PFA in PEM buffer (80 mM PIPES pH 6.8, 5 mM EGTA, 2 mM MgCl2) for 10 min at room temperature (RT). After rinses in 0.1 M phosphate buffer (PB), neurons were blocked for 2–3 h at RT in immunocytochemistry buffer (ICC: 0.22% gelatin, 0.1% Triton X-100 in PB), and incubated with primary antibodies diluted in ICC overnight at 4 °C. After rinses in ICC, neurons were incubated with secondary antibodies diluted in ICC for 1 h at RT, rinsed and incubated in fluorescent phalloidin at 0.5 μM for either 1 h at RT or overnight at 4 °C. Stained coverslips were kept in PB + 0.02% sodium azide at 4 °C before SMLM imaging. For epifluorescence imaging, coverslips were mounted in ProLong Glass (Thermo Fisher #P36980).

**Unroofing and PREM immunocytochemistry**. Unroofing was performed by sonication as previously described[18]. Coverslips were quickly rinsed three times in Ringer + Ca (155 mM NaCl, 3 mM KCl, 3 mM NaH₂PO₄, 5 mM HEPES, 10 mM glucose, 2 mM CaCl₂, 1 mM MgCl₂, pH 7.2), then immersed 10 s in Ringer-Ca (155 mM NaCl, 3 mM KCl, 3 mM NaH₂PO₄, 5 mM HEPES, 10 mM glucose, 3 mM EGTA, 5 mM MgCl₂, pH 7.2) containing 0.5 mg/mL poly-L-lysine, then quickly rinsed in Ringer-Ca then unroofed by scanning the coverslip with rapid (2–5 s) sonicator pulses at the lowest deliverable power in KHMgE buffer (70 mM KCl, 30 mM HEPES, 5 mM MgCl2, 3 mM EGTA, pH 7.2).

Unroofed cells were immediately fixed in KHMgE: 4% PFA for 10 min for epifluorescence or SMLM of fluorescence-labeled samples, 4% PFA for 45 min for PREM of immunogold-labeled samples, 3% PFA–1% glutaraldehyde or 2% PFA–2% glutaraldehyde for 10–20 min for PREM and correlative SMLM/PREM. Glutaraldehyde-fixed samples were subsequently quenched using 0.1% NaBH₄ in KHMgE for 10 min. Immunofluorescence labeling was performed as above, but replacing the ICC buffer with a detergent-free buffer (KHMgE, 1% BSA). Immunogold labeling was performed in the same detergent-free solution: samples were blocked for 30′, incubated 1 h 30 min with the primary antibodies diluted to 1:20, rinsed, incubated two times 20 min with the gold-coupled secondary antibodies, then rinsed.

**Actin immunogold and myosin S1 labeling**. For immunogold labeling of actin[16], unroofed neurons were fixed with 2% PFA in KHMgE, then quenched for 10 min in KHMgE, 100 mM glycine, 100 mM NH₄Cl. After blocking in KHMgE, 1% BSA, they were incubated with phalloidin-Alexa Fluor 488 (0.5 μM) for 45 min, then immunolabeled using an anti-Alexa Fluor 488 primary antibody and a gold-

coupled goat anti-rabbit secondary antibody as described above for immunogold labeling.

For myosin S1 labeling, neurons were unroofed in PEM100 buffer (100 mM PIPES-KOH pH 6.9, 1 mM MgCl2, 1 mM EGTA), treated with 0.25 mg/mL myosin S1 in PEM100 buffer for 30 min, then fixed with 2% glutaraldehyde in PEM100 buffer for 10 min.

**Platinum-replica sample processing**. Fixed samples were stored and shipped in KHMgE, 2% glutaraldehyde. Cells were further sequentially treated with 0.5% OsO4, 1% tannic acid and 1% uranyl acetate prior to graded ethanol dehydration and hexamethyldisilazane substitution (Sigma). Dried samples were then rotary-shadowed with 2 nm of platinum and 5–8 nm of carbon using an ACE600 high-vacuum metal coater (Leica Microsystems). The resultant platinum-replica was floated off the glass with hydrofluoric acid (5%), washed several times on distilled water, and picked up on 200 mesh formvar/carbon-coated EM grids.

**Epifluorescence and SMLM**. Diffraction-limited images were obtained using an Axio-Observer upright microscope (Zeiss) equipped with a ×40 NA 1.4 or ×63 NA 1.46 objective and an Orca-Flash4.0 camera (Hamamatsu). Appropriate hard-coated filters and dichroic mirrors were used for each fluorophore. Quantifications were performed on raw, unprocessed images. An Apotome optical sectioning module (Zeiss) and post-acquisition deconvolution (Zen software, Zeiss) were used to acquire and process images used for illustration.

For single-color SMLM, we used STochastic Optical Microscopy (STORM). STORM was performed on an N-STORM microscope (Nikon Instruments). Coverslips were mounted in a silicone chamber filled with STORM buffer (Smart Buffer Kit, Abbelight). The N-STORM system uses an Agilent MLC-400B laser launch with 405 nm (50 mW maximum fiber output power), 488 nm (80 mW), 561 mW (80 mW), and 647 nm (125 mW) solid-state lasers, a ×100 NA 1.49 objective and an Ixon DU-897 camera (Andor). After locating a suitable neuron using low-intensity illumination, a TIRF image was acquired, followed by a STORM acquisition. 30,000–60,000 images (256 × 256 pixels, 15 ms exposure time) were acquired at full 647 nm laser power. Reactivation of fluorophores was performed during acquisition by increasing illumination with the 405 nm laser. When imaging actin, 30 nM phalloidin-Alexa Fluor 647 was added to the STORM buffer to mitigate actin unbinding during imaging[49].

For two-color SMLM (Fig. 1d, e), we used STORM in combination with DNA-PAINT[47]. Neurons were labeled using secondary antibodies coupled to a PAINT DNA handle for β4- or β2-spectrin (rabbit P3 or mouse P1, respectively), as well as phalloidin-Alexa Fluor 647 for actin. Imaging was done sequentially[49], first for actin in STORM buffer (60,000 frames at 67 Hz), then for β4- or β2-spectrin in PAINT buffer (0.1 M phosphate buffer saline, 500 mM NaCl, 5% dextran sulfate, pH 7.2) supplemented with 0.12–0.25 nM of the corresponding PAINT imager strand coupled to Atto650 (Metabion, 40,000 frames at 33 Hz). No chromatic aberration correction was necessary as both channels were acquired using the same spectral channel, and translational shift was corrected by autocorrelation and manual refinement.

The N-STORM software (Nikon Instruments) was used for the localization of single fluorophore activations. After filtering out localizations to reject too low and too high photon counts, the list of localizations was exported as a text file. Image reconstructions were performed using the ThunderSTORM ImageJ plugin[50] in Fiji software[51]. Custom scripts and macros were used to translate localization files from N-STORM to ThunderSTORM formats, as well as automate images reconstruction for whole images and detailed zooms.

**Fluorescence image analysis**. Quantification of the labeling intensities on epifluorescence images was done by first tracing the region of interest along a process (AIS, axon or dendrite) using the NeuronJ plugin[52] in Fiji software. Tracings were then translated into ImageJ ROIs and the background-corrected intensities were measured for each labeled channel.

Quantification of the scaffold periodicity on SMLM images was performed using autocorrelation analysis:[7] if an intensity profile is periodically patterned, its autocorrelation curve will exhibit regular peaks at multiples values of the period (here 190 nm, 380 nm, 760 nm etc). The position of the first peak of the autocorrelation curve can be used to retrieve the average spacing (s), and its height estimates how marked the periodicity of the profile is. Axons were manually traced in ImageJ using polyline ROIs on 16 nm/pixel reconstructed images. The normalized autocorrelation curve of the corresponding intensity profile was calculated and plotted. Autocorrelation curves of all tracings for a given labeling and condition were then averaged. The first non-zero peak of the averaged autocorrelation curve was fitted in Prism (Graphpad software) to estimate its position, providing the spacing values and the corresponding error. The autocorrelation amplitude (height of the first peak) was estimated from the difference between the autocorrelation values at 192 nm (approximate position of the first peak) and 96 nm (approximate position of the first valley).

**EM of platinum replicas**. Replicas on EM grids were mounted in a eucentric side-entry goniometer stage of a transmission electron microscope operated at 80 kV (model CM120; Philips) and images were recorded with a Morada digital camera

(Olympus). Images were processed in Photoshop (Adobe) to adjust brightness and contrast and presented in inverted contrast. Tomograms were made by collecting images at the tilt angles up to ± 25° relative to the plane of the sample with 5° increments. Images were aligned by layering them on top of each other in Photoshop. Measurement of actin braids spacing, length and width were performed on high-magnification PREM views using ImageJ. Five lines were traced between consecutive braids (for braid spacing) or across a single braid, before and after eventual splitting (for braid width) and averaged to obtain each distance data point.

**Correlative SMLM/PREM**. For correlative SMLM/PREM, unroofed and stained samples were first imaged by epifluorescence, then a STORM image was acquired (see above). At the end of the acquisition a objective-style diamond scriber (Leica) was used to engrave a ~1 mm circle around the imaged area, and a low-magnification, ×10 phase image was taken as a reference for relocation. After replica generation and prior to its release, the coated surface of each coverslip was scratched with a needle to make an EM grid sized region of interest containing the engraved circle. Grids were imaged with low-magnification EM to relocate the region that previously imaged by epifluorescence and SMLM, and high-magnification EM views were taken from the corresponding axonal region. A high-resolution SMLM reconstruction was mapped and aligned by affine transformation to the corresponding high-magnification EM view using the eC-CLEM plugin in ICY software[53].

**Data representation and statistical analysis**. Intensity profiles, graphs and statistical analyses were generated using Prism. On bar graphs, dots are individual measurements, bars or horizontal lines represent the mean, and vertical lines are the SEM unless otherwise specified. Significances were tested using two-tailed unpaired non-parametric $t$ tests (two conditions) or one-way non-parametric ANOVA followed by Tukey post-test (three or more conditions). In all figures, significance is coded as: ns non-significant, $*p < 0.05$, $**p < 0.01$, $***p < 0.001$. The number of data points and experiments for quantitative results is presented in Supplementary Tables 1 and 2. The measurements presented in Fig. 1 are from the same pooled dataset as the control condition in Fig. 3. Statistics from quantitative analyses are summarized in Supplementary Tables 1–3.

**Reporting summary**. Further information on research design is available in the Nature Research Reporting Summary linked to this article.

## Data availability

Data supporting the findings of this manuscript are available from the corresponding authors upon reasonable request. A reporting summary for this Article is available as a Supplementary Information file. The source data from Figs. 1f–h, l, 2c, f, and Supplementary Fig. 3b–f are available as Source Data File. All data shown in the figures (graph data, native resolution PREM images, SMLM localization files and reconstructions, fluorescence images) are available as a compressed archive file on FigShare at https://doi.org/10.6084/m9.figshare.10261124.

## Code availability

Code for data analysis is available on GitHub: STORM processing scripts used in this work can be found at https://github.com/cleterrier/ChriSTORM. The ImageJ script for generating autocorrelation curves from polyline ROIs is available at https://github.com/cleterrier/Process_Profiles. Intensity measurement scripts are available at https://github.com/cleterrier/Measure_ROIs.

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

## Acknowledgements

We thank Matthew Rasband and Vann Bennett for antibody gifts; Fanny Boroni-Rueda for help with neuronal cultures; the NCIS imaging facility and Nikon Instruments for SMLM equipment; the IBPS cryo-EM facility for EM equipment; Subhojit Roy, Jeanne Lainé, Marc Bitoun, Ricardo Henriques, Manuel Théry, Laurent Blanchoin, Emmanuel Nivet, and the NeuroCyto team members for discussions and critical reading of the manuscript. This work has been funded by the CNRS ATIP-AVENIR program (grant ATIP AO 2016 to CL); by Sorbonne Université, INSERM, Association Institut de Myologie core funding and the Agence Nationale de la Recherche (grant ANR14-CE12-0001-01 to SV).

## Author contributions

S.V.: conceptualization, formal analysis, funding acquisition, investigation, methodology, visualization, writing–review and editing. S.G., A.J., and G.C.: formal analysis, investigation. C.L.: conceptualization, formal analysis, funding acquisition, investigation, methodology, visualization, software, supervision, writing—original draft, review, and editing.

## Competing interests

The authors declare no competing interests
