## [Peer Review File · Nature Communications]

Reviewers' Comments:

Reviewer #1:

Remarks to the Author:

This is collaborative work from an expert group in platinum replica electron microscopy (PREM) of the actin cytoskeleton with a group focused on the organization of the axonal cytoskeleton using single molecule localization-based microscopy (SMLM) approaches.

The authors establish an assay for the correlative investigation of the recently discovered, spectrin interconnected axonal actin rings in ultrasound-unroofed cells via SMLM and PREM. They then perform pharmacological treatments to influence actin dynamics to ask, whether the actin structures undergo detectable changes in organization.

They find that the putative actin rings as verified by correlative SMLM and PREM consist of two intertwined actin filaments and describe the interconnection by spectrin beta isoforms 2 and 4.

The data are presented in an excellent fashion and allow access to both EM data and correlated EM-SMLM data in parallel, a very well-documented and excellently executed verification of the experimental system and convincing identification of the respective components.

The authors find that unroofing of cells via short ultrasound pulses leaves the ventral membrane cytoskeleton in the initial axon intact as detectable and quantified by the periodicity of actin and spectrin isoforms.

In the reviewers opinion, the data support the notion that the actin structures the authors detect are indeed at least remnants of the axonal actin rings that have by now also detected in live-cell microscopy.

The manuscript seems very short, so there is ample space to discuss the author's findings in the light of previous work by the Svitkina laboratory. It is clear to me, why the results presented here look very different from the Jones et al paper, but the authors should at least state clearly in the discussion why that is. Do the authors see any ion channels or AnkyrinG? A DIV 13 – 17, many ion channels and AnkyrinG should be observable in the AIS.

Overall, I find this is an excellent, timely and important study that would find great resonance in the community if it would provide a more complete picture. The authors provide a much anticipated first electron microscopy look at the axonal actin rings that were not detected by earlier studies but unfortunately stop there. The proof of principle is excellently executed, and the reader hopes for a complete study of the AIS and axonal cytoskeletal structure, but is left somewhat disappointed. As the pipeline for correlative microscopy is established I would consider this an "accept with minor changes" and experiments on a few components should be doable in a few weeks.

Major points:

- As it is the scope of this manuscript is quite narrow and a lot of obvious questions remain unanswered, the answering of which would greatly increase the value of this manuscript as a resource to the community.
- Such longer actin filaments are more suited for myosin motors to interact with and pull together than short bundles. One of the authors has recently shown that phosphorylated myosin light chain would localize to actin rings in the AIS (Berger et al., Neuron 2018). Can the authors detect myosin or phosphorylated myosin in their preparations and where would it localize/be oriented?
- Can the authors detect any ion channels or ankyrin isoforms? In the previous PREM study of the AIS (Jones et al., 2014), such features were prominently visible in the AIS at the developmental stage the authors use in the present study. Or are the Ion channels covered by the spectrin meshwork? Where is adducin? To all these molecules excellent antibodies exist that have been established in SMLM by the authors.
- As the authors have previously discovered a connection between microtubules EB3 and AnkyrinG, is there anything the authors can conclude towards the connection between microtubules and the

AIS organization?

Minor points:

- This is the first description of the axonal actin rings by electron microscopy and the acquired images are a great resource for the community. The authors should make large highest resolution images available as supplementary data for the reader to appreciate and as a resource in the field. While in the current form the compression for the pdfs excellent and allows to zoom in to high resolution, it is doubtful this will be possible in the final online pdf version in the published paper.
- The manuscript is quite short, likely because it was transferred from a journal with rigorous space constraints? Anyway, it would profit from an expanded description and discussion of related work, especially the differences observed in contrast to the Jones et al (JCB 2014) study and possible implications regarding the fixation method and accessibility/visibility in the alternative approaches.

Reviewer #2:

Remarks to the Author:

This study examines periodic actin rings in axons of cultured neurons that have had their plasma membranes mechanically removed using both super-resolution light microscopy and platinum replica EM as well as correlative super-resolution light/platinum replica EM. The approach of mechanically detaching the cortical plasma membrane was introduced by John Heuser and allows for better preservation of internal cellular detail than current approaches of detergent extraction, where the plasma membrane collapses and can obscure high resolution imaging. An interesting new finding here is that the actin filaments are actually two filaments that entwined or "braided". I am not aware that this configuration of actin filaments has been reported before. I took the liberty of consulting with Tom Pollard who is highly informed regarding actin. He of course knew about actin filament bundles but not just two actin filaments. So, I believe that this is a "first". As the authors point out, such braided filaments would be more stable and resist depolymerization. Missing from the study is a mechanism for actin filament association. It would be of considerable interest if the authors could examine adducin-deficient axons, as adducin could be the "entwining" factor. However, this experiment, in my view, is not required for publication as a Nature Communication. Overall, this is a carefully executed and well written paper. I have only a minor comment.

1. It would be helpful to further support the two-filament model to include histograms of the widths of the "braided" filaments, and the individual filaments that diverge at "y" junctions, as well as microtubules.

Vann Bennett

Reviewer #3:

Remarks to the Author:

To date, a controversy in the field of cellular neurobiology is the structural organization of the cytoskeleton in axons. Specifically, a landmark 2013 paper used super-resolution imaging to reveal a system of dramatic periodic actin/spectrin rings that assembled in axons. The rings were proposed to mechanically stabilize the axon and organize channels and other receptors along the axon shaft. However, other work using more classic EM methods from Jones et al. was unable to see this periodic actin assembly and instead argued for a more dense, compact actin meshwork in axons. Currently, it is unclear how the two arrangements for actin obtained by these different nanoscopic imaging methods relate to the true structure of the cytoskeleton in living axons. In this paper by Vassilopoulos et al., this question is revisited with metal replica EM, super-resolution imaging, and correlative super-resolution light and EM. Through a powerful set of elegant

experiments, the authors present a new actin structure in the axon that appears to be composed of submicron links of two twisted actin fibers that are rimmed by perpendicular septin fibers. The molecular identity of these elements are confirmed by immunogold EM in metal replicas. The authors go on to show that these structures can be disrupted by drugs. Finally, the authors argue with correlative methods that the stripes of actin observed in super-resolution fluorescence correspond to these actin fibers. In general, I think this paper is extremely interesting, with beautiful images, and thoughtful and careful analysis and experiments. It is a nice addition to the field of neuronal cell biology and will act as a catalyst for future work into the cytoskeleton and other structures found in neurons. I do, however, have several comments and concerns that should be addressed in a future version of the manuscript and would substantially improve the work. My specific comments follow.

1. The paper's primary finding is that these unique actin filaments are stretched perpendicular to the axon and form relatively long "twisted" fibers that sometime splay into a "Y" shape. These fibers are arranged at an interval of around ~200 nm which nicely fits with the super-resolution spacing. However, because of a lack of fine-scale detail in the EM images, I cannot see the "twisting" or evidence that there are more than one actin fiber in these filaments. Finally, the splaying at the ends could be branched filaments that contact the primary filament; a common arrangement in the actin cortex. Why do the authors think these filaments are braided or twisted? Because this is such a unique structure for actin (the authors state "this organization of actin as braids of long, unbranched filaments is to our knowledge unique and contrasts with the currently assumed view that axonal actin rings made of short, capped and bundled actin filaments") the reader should have confidence in this new finding. From my eye, I cannot unmistakably see this structure. Are the filaments twice as wide as expected? Is there a regular expected braid twist seen along the filament that can be measured and isn't present in the other actin filaments in the same sample? This analysis would control for uneven metal coating and drying issues. I think a bit more is needed here to make for a strong argument for a totally new "twisted" fiber arrangement. This is such a key and novel part of the paper it would help to strengthen the evidence.

2. The correlative work, while impressive, is missing most of the unique actin structures. Thus, I am not sure how much this figure is adding to the story. It appears that the key actin filaments were lost or destroyed during the fluorescence imaging or maybe didn't survive the shipping for EM??Regardless, the remaining material left after the fluorescence imaging is hardly visible in EM. I can only see a few matching filaments in these images. Some of the images have no filaments that correspond to the fluorescence signal. I am not sure how this could be quantitatively analyzed to support the author's argument but it might be nice to provide some statistical measurement of how often filaments corresponded to the fluorescence or alternatively how often EM filament had fluorescence. As it stands now, I don't feel like this figure is strongly supporting the main arguments of the paper over and above what is shown in the super-resolution alone, EM alone, and immunogold EM images alone. Technically, this figure is an amazing achievement but the data just don't look that convincing.

3. It would be nice to provide images of the immuno-gold that did not have color augmentation.

4. It would be nice to always have corresponding images in the EM figures that do not have color augmentation so that the reader can evaluate the structures without the input of the authors.

5. How many cells were imaged for EM? This information is not present as far as I can tell in the document. How often were these actin fibers seen? Was it common or rare? This is important information to put in the main text or figure legend for the reader. I would like to see an "n" for number of cells, number of EM preps, and number of structures observed. This information was given for the fluorescent data but not EM or CLEM.

Response to reviewers for “Ultrastructure of the axonal periodic scaffold reveals a braid-like organization of actin rings”, manuscript NCOMMS-19-20620-T

S. Vassilopoulos, S. Gibaud, A. Jimenez, G. Caillol, C. Leterrier

We would like to thank the three Reviewers for the detailed assessment of our manuscript and helpful remarks and suggestions. Based on the feedback from them and from the Editors, we have substantially revised our manuscript to address the points raised. As a result, the revised manuscript contains a number of new experimental results and analysis that we hope will significantly both strengthen and broaden its conclusions.

We provide new super-resolution (SMLM), immunogold PREM and correlative SMLM/PREM data for additional AIS components, as requested by Reviewer #1. We resolved the organization of phospho-myosin light chain (pMLC) that is associated in the AIS actin/spectrin scaffold, and obtained new insights into the connection between the scaffold and microtubules by examining the ankyrin G tail localization. We have extensively quantified the thickness of axonal actin braids to show that they are made of two actin filaments, as suggested by Reviewer #2 and Reviewer #3. We also addressed Reviewer #3 concerns about the relationship between SMLM and PREM data in the correlative experiments. In addition to the correlative results on additional MPS components, we acquired new data for correlative SMLM/PREM imaging of actin and β 4-spectrin.

Beyond this summary of the most important changes, please find below the detailed responses to each of the Reviewer’s points and how we addressed them in this revised version of our manuscript. Throughout this response, Reviewer comments are italicized and highlighted in light blue; citation from our revised manuscript are italicized. In the revised manuscript text document, all modifications relatively to the initial version are highlighted in light grey. Please excuse the length of some replies as we wanted to be as precise and extensive as possible in our answers.

Reviewer #1

This is collaborative work from an expert group in platinum replica electron microscopy (PREM) of the actin cytoskeleton with a group focused on the organization of the axonal cytoskeleton using single molecule localization-based microscopy (SMLM) approaches. The authors establish an assay for the correlative investigation of the recently discovered, spectrin interconnected axonal actin rings in ultrasound-unroofed cells via SMLM and PREM. They then perform pharmacological treatments to influence actin dynamics to ask, whether the actin structures undergo detectable changes in organization. They find that the putative actin rings as verified by correlative SMLM and PREM consist of two intertwined actin filaments and describe the interconnection by spectrin beta isoforms 2 and 4. The data are presented in an excellent fashion and allow access to both EM data and correlated EM-SMLM data in parallel, a very well-documented and excellently executed verification of the experimental system and convincing identification of the respective components. The authors find that unroofing of cells via short ultrasound pulses leaves the ventral membrane cytoskeleton in the initial axon intact as detectable and quantified by the periodicity of actin and spectrin isoforms. In the reviewer’s opinion, the data support the notion that the actin structures the authors detect are indeed at least remnants of the axonal actin rings that have by now also detected in live-cell microscopy.

We would like to thank Reviewer #1 for the appreciation of our work, in particular the correlative SMLM/PREM data experiments and for deeming our overall conclusions convincing.

The manuscript seems very short, so there is ample space to discuss the author's findings in the light of previous work by the Svitkina laboratory. It is clear to me, why the results presented here look very different from the Jones et al paper, but the authors should at least state clearly in the discussion why that is. Do the authors see any ion channels or AnkyrinG? A DIV 13 –17, many ion channels and AnkyrinG should be observable in the AIS.

These are important points that constitute **minor point 1.5** and **major point 1.2** below. We have added a short discussion on why our results differ from previous work of the Svitkina lab (Jones et al., 2014), and have also performed experiments to visualize ankyrin G and Nav channel - please refer to our detailed responses below.

Overall, I find this is an excellent, timely and important study that would find great resonance in the community if it would provide a more complete picture. The authors provide a much anticipated first electron microscopy look at the axonal actin rings that were not detected by earlier studies but unfortunately stop there. The proof of principle is excellently executed, and the reader hopes for a complete study of the AIS and axonal cytoskeletal structure, but is left somewhat disappointed. As the pipeline for correlative microscopy is established I would consider this an “accept with minor changes” and experiments on a few components should be doable in a few weeks.

Major points:

As it is the scope of this manuscript is quite narrow and a lot of obvious questions remain unanswered, the answering of which would greatly increase the value of this manuscript as a resource to the community.

1.1 - *Such longer actin filaments are more suited for myosin motors to interact with and pull together than short bundles. One of the authors has recently shown that phosphorylated myosin light chain would localize to actin rings in the AIS (Berger et al., Neuron 2018). Can the authors detect myosin or phosphorylated myosin in their preparations and where would it localize/be oriented?*

We thank the reviewer for this suggestion. Indeed, the organization of myosins in the proximal axon is a timely question, following the recent report from Berger et al. that found phospho-myosin light chain (pMLC) concentrated at the AIS and apposed to actin rings (Berger et al., 2018).

A side note about immunolabeling in the context of our experiments, that is relevant to myosins and all components of the MPS: after unroofing, immunolabeling is done in the absence of any detergent or permeabilizing agent in order to keep the ultrastructure intact for PREM. For this reason, the efficiency of the immunolabeling can differ significantly from what is obtained after regular fixation and permeabilization, and a “good” antibody can result in very little to no labeling. Two likely factors are at play: soluble proteins that are not tightly bound to the cytoskeleton are washed out during the mechanical unroofing, and proteins that are deeply embedded in the scaffold can have their epitope inaccessible in the absence of detergent. The lower efficiency of immunogold labeling (likely due to the additional size of the gold beads conjugated to secondary antibodies) is a further constrain. Our tests showed that we could successfully label some components of the MPS after unroofing (such as pMLC), but not others (see the case of Nav channels and ankyrin G below).

As regards the suggestion of localizing myosins, we tested and optimized pMLC labeling on unroofed neurons and could obtain SMLM, immunogold PREM and correlative SMLM/PREM images. These

experiments are now presented in the Results section, **p. 3-4 lines 92-101, Fig. 3a-e** for SMLM and immunogold PREM, and **p. 6 lines 165-169, Supplementary Fig. 6** for correlative SMLM/PREM. Despite a partial loss of staining after unroofing (compare intact and unroofed neurons on the widefield images of **Fig. 3f vs 3h**, as well as on the SMLM images of **Fig. 3g vs 3i**), we could visualize examples of pMLC immunogold and correlative fluorescence appearing on filaments thicker than actin braids, and oriented perpendicular or diagonally relatively to the braids (brackets on **Fig. 3j and Supplementary Fig. 6c**). These filaments/rods could be myosins. We did not obtain a good staining using non-muscle myosin II antibodies that usually works well for regular immunolabeling. We think that our result brings insight to the debate of myosins organization in the axon, and the possibility that they could crosslink adjacent rings (as we mention **p. 4 line 100-101**). This is particularly timely in the context of recent preprints studying the organization of myosins and their partners in the axon, and how they can drive radial and/or longitudinal tension (Wang et al., 2018; Abouelezz et al., 2019). This point is now mentioned in the revised Discussion section, p. 6-7 lines 191-194: “Further studies are needed to clarify the precise localization of adducin as well as to confirm the localization of myosin filaments associated with the MPS, in order to explain how actin-associated proteins can regulate radial and longitudinal tension along the axon as well as axonal diameter.”

1.2 - Can the authors detect any ion channels or ankyrin isoforms? In the previous PREM study of the AIS (Jones et al., 2014), such features were prominently visible in the AIS at the developmental stage the authors use in the present study. Or are the Ion channels covered by the spectrin meshwork? Where is adducin? To all these molecules excellent antibodies exist that have been established in SMLM by the authors.

We would like to thank the reviewer for this suggestion of extending our results to AIS-specific components of the MPS, in line with our previous studies (Letierrier et al., 2015). We focused on several key components of the AIS during the course of this revision:

Nav channels. We have tested antibodies to label Nav channels and ankyrin G after unroofing. We could not get any staining for Nav channels using the reference anti pan-Nav antibody (Sigma clone K58/35): the antibody works very well after fixation and permeabilization (**Reviewer Figure 1a**), but no staining remains after unroofing and immunolabeling without permeabilization, either using paraformaldehyde (**Reviewer Figure 1b**) or a mix of paraformaldehyde and glutaraldehyde for fixation (**Reviewer Figure 1c**). We think this results from the low accessibility of the channels intracellular epitope (located within the loop I-II) that is likely to be deeply embedded within the spectrin mesh. We thus did not use this antibody further.

Reviewer Figure 1. Nav channel labeling of intact and unroofed neurons (a-c) Epifluorescence image of a neuron labeled for actin (gray), $\beta 4$ -spectrin (orange) and Nav channels (blue, Sigma anti pan-Nav K58/35): intact neuron after PFA fixation and permeabilization (a), unroofed neuron after PFA (b) and PFA/gluta (c) fixation without permeabilization. The Nav staining is absent in unroofed neurons. Scale bars 20 μ m.

Ankyrin G. For ankyrin G, we tested two antibodies: a monoclonal (NeuroMab clone 106/65) for which we previously mapped the epitope to the spectrin-binding domain of ankyrin (ankG SB antibody), close to the membrane-bound aminoterminal ankyrin repeats (Leterrier et al., 2015) and a polyclonal against a 200-aminoacid stretch of the 480 kDa ankyrin G tail (ankG 480 tail antibody), kindly provided by Vann Bennett (Jenkins et al., 2015). The anti-ankyrin G SB antibody was working well for classical labeling (Reviewer Figure 2a) but resulted in a faint labeling after unroofing (Reviewer Figure 2b-c). By contrast, the ankG 480 tail antibody provided an intense staining both in intact and unroofed conditions, and we chose to use it for further experiments (SMLM, immunogold PREM, correlative SMLM/PREM). The results using this ankG 480 tail antibody are now part of the revised manuscript, p. 4 lines 102-114, Fig. 3f-j for widefield, SMLM and immunogold PREM, and p. 6 lines 170-177, Fig. 6d-f for correlative SMLM/PREM. The ankG 480 tail labeling was very dense, with gold beads covering the MPS mesh in immunogold PREM experiments (Fig. 3j). We think that because it is a polyclonal antibody against a large protein fragment (200 aminoacids), it is likely that it contains IgGs capable

Reviewer Figure 2. Labeling of intact and unroofed neurons for ankyrin G spectrin-binding domain (a-c) Epifluorescence image of a neuron labeled for actin (gray) and ankyrin G spectrin-binding domain (orange, ankG SB, NeuroMab clone 106/65): intact neuron after PFA fixation and permeabilization (a), unroofed neuron after PFA (b) and PFA/gluta (c) fixation without permeabilization. The ankG SB staining is faint in unroofed neurons. Scale bars 20 μ m.

of

Reviewer Figure 3: Molecular structure of ankyrin G and AIS spectrins (adapted from Leterrier, 2018).

(a) Domain organization of ankyrin G, which exists as isoforms of 480 and 270 kDa at the AIS (top, orange). Antibodies described in the text (red) and EB-binding SxIP motifs (blue) are indicated. Binding sites of partners are indicated below the protein (bars). Lengths are to scale with the aminoacid lengths. (b) Domain organization of β 4-spectrin, which exists as isoforms of 280 and 140 kDa (left), and domain organization of α 2-spectrin (right). Binding sites of partners are indicated below the protein (grey bars). Bottom, structure of the α 2/ β 4 spectrin tetramer (N and C refer to the aminoterminal and carboxyterminus of each subunit). (c) The AIS submembrane complex. The α 2/ β 4 spectrin tetramers (red) lie horizontally under the plasma membrane (dark grey), connecting actin rings (purple) with a distance of ~190 nm. In the middle of the tetramer, ankyrin G (orange) is bound to β 4-spectrin and anchors AIS membrane proteins (Nav channels, CAMs, blue).

epitopes, explaining the dense staining obtained (see **Reviewer Figure 3** for MPS proteins domains, localization of antibodies and tridimensional arrangement). In some instances, we could visualize binding of two gold beads in-between actin rings along a spectrin rod (**Fig. 3j**, brackets), which is consistent with our current model of the AIS organization: spectrin tetramers connect actin rings, with ankyrin G occupying two binding sites on β 4-spectrin close to the center of the tetramer (**Reviewer Figure 3c**). However, the mesh was studded with gold beads, in line with the idea that the long tail of ankyrin G can extend significantly from the binding site to β 4-spectrin (Letierrier et al., 2015). Another advantage of using this antibody against the ankG 480 tail is that it contains most of the SxIP motifs responsible for binding to End-Binding (EB) proteins and mediate the interaction between ankyrin G and microtubules (Letierrier et al., 2011; Fréal et al., 2016), allowing us to address the connection of the MPS to microtubules (see **major point 1.3** below).

Adducin. Our results showing that rings are made of long, intertwined filaments lead to reconsider the current model of actin rings made of short filaments, capped by adducin. This is an important aspect of our work, and we developed this more explicitly in the revised Discussion section **p. 6 lines 187-193**: “*We show that axonal actin rings are made of a small number of long, intertwined filaments rather than by a large number of short, bundled filaments as previously hypothesized. This short filament model was inferred from the presence of adducin at actin rings, at least along the distal axon, because adducin was described as a capping protein that can bind to the barbed end of filaments. However, adducin is also able to associate to the side of filaments, enhancing the lateral binding of spectrin to actin. [...] Our results suggest that this lateral binding role is dominant to enhance actin rings interaction with spectrins*”. The localization and role of adducin are thus relevant questions, as also noted by Reviewer #2. We devoted a significant effort to visualize adducin after unroofing by immunofluorescence, SMLM and immunogold PREM. However, we could not obtain images of sufficient quality using the reference antibodies previously validated by SMLM (abcam ab51130, (Xu et al., 2013).

We think the main reason here is that adducin is primarily present along the distal axon, and much less along the AIS and proximal axon, the portion of the axon that is reliably unroofed (see **p. 2 line 44**: “*it was more difficult to unroof distal processes*”). This was reported previously (Jones et al., 2014) and we indeed found the proximal axon to be largely devoid of adducin staining. The overall

Reviewer Figure 4. Low presence of adducin along the proximal axon of intact and unroofed neurons

(a-b) Epifluorescence images of a mature neuron labeled for map2 (blue), β 2-spectrin (red) and adducin (green). Zooms show the low staining of adducin and β 2-spectrin along the proximal axon/AIS (bracket). (c) Epifluorescence image of a neuron labeled for ankyrin G (ankG, magenta) and adducin (green). The adducin labeling is very faint along the unroofed portion of the axon. (d) SMLM image of the proximal axon of an unroofed neuron labeled for adducin. The low amount of adducin results in very sparse clusters on the SMLM image. Scale bars 20 μ m (a-c), 2 μ m (d, left), 0.5 μ m (d, right).

profile of adducin along the axon is similar to that of β 2-spectrin, with a low abundance at the AIS and a presence along the distal axon (**Reviewer Figure 4a-b**) (Galiano et al., 2012). In addition, the adducin labeling does not work well after the unroofing and labeling without permeabilization, as seen on widefield and SMLM images (**Reviewer Figure 4c-d**). We obtained images of β 2-spectrin labeling by immunogold PREM and correlative SMLM / PREM, albeit not very informative ones due to the low density of staining (**Supplementary Figures 2 and 5**). We did not manage to obtain satisfying images to correlate adducin labeling with the underlying structure. We now mention this in the revised text, **p.3 lines 83-86**: “*The immunolabeling efficiency along the unroofed proximal axons was low for β 2-spectrin, as it is mostly present along the distal axon (Supplementary Figure 2a). Similarly, we could not reliably detect adducin, as it is mostly absent from the AIS and proximal axon.*” and **p. 6 lines 191-192**: “*Although we were not able to directly localize adducin at the ultrastructural level, our results suggest...*”. We hope that future studies will clarify the localization and role of adducin, as well as identify potential proteins playing a similar role in the AIS, as we now mention in the revised Discussion section **p.6 line 193**: “*Further studies are needed to clarify the precise localization and role of adducin*”.

1.3 - *As the authors have previously discovered a connection between microtubules EB3 and AnkyrinG, is there anything the authors can conclude towards the connection between microtubules and the AIS organization?*

We thank the reviewer for this appreciation of our earlier work. Indeed, we previously showed that ankyrin G links the MPS (via its interaction with β 4-spectrin and membrane proteins) to microtubules via EB1/EB3 proteins (Leterrier et al., 2011), a finding confirmed and extended since, delineating the importance of this link for AIS assembly and maintenance (Fréal et al., 2016; 2019). As our work was not focusing on AIS-specific components, we initially did not investigate this connection in our PREM data in detail. It is clear that regularly spaced braids in proximal axons are often observed along portions of unroofed membrane with microtubules running along them; the braids even seem to connect perpendicularly to microtubules in some instances (**Fig. 1j** and **Supplementary Movie 1**). In fact, the braids are running under the microtubules: correlative SMLM/PREM reveals that they usually continue on the other side of microtubules, embedded in the spectrin mesh (**Fig. 5c**). We now mention this hint of a relationship between actin braids/MPS and the microtubules at the end of the revised Results section, **p. 6 lines 176-177**: “*we usually detected well-preserved actin braids along unroofed areas where microtubule bundles are still present (see Fig. 1j-k).*”

Thanks to the Reviewer #1 suggestion, our new data in the ultrastructural localization of the ankG 480 tail now provides further insight into this connection between the MPS and microtubules via ankyrin G. On SMLM images of unroofed samples, we could clearly visualize longitudinal gaps flanked by a high intensity of ankyrin G staining, a pattern we refer to as “tracks” in the revised manuscript (**p.4 line 108, Fig. 3i**). We come back to this initial observation when presenting the correlative SMLM/PREM data: correlative imaging allows us to show that these tracks correspond to microtubules. The microtubules themselves are devoid of staining, and globular material attached to them correspond to bright clusters of ankyrin G tail labeling on the SMLM image (**Fig. 6f**). It is likely that what we visualized here are the contacts between ankyrin G tails and microtubules via EB proteins, as we mention in the revised Results section, **p.6 lines 175-176**: “*This is the first direct visualization of the MPS association with microtubules via ankG along the AIS*”. We think this is a significant observation and thank Reviewer #1 for suggesting these experiments.

Minor points:

1.4 - *This is the first description of the axonal actin rings by electron microscopy and the acquired images are a great resource for the community. The authors should make large highest resolution images available as supplementary data for the reader to appreciate and as a resource in the field. While in the current form the compression for the pdfs excellent and allows to zoom in to high resolution, it is doubtful this will be possible in the final online pdf version in the published paper.*

We would like to thank Reviewer #1 for highlighting the novelty of our work. We have tried to keep the image quality in the current manuscript as high as possible with a still reasonable file size by optimizing the PDF export parameters, and we thank Reviewer #1 for appreciating this effort. Even at the current quality, the Figures can only show a portion of the original image details, in particular for the PREM views. To allow readers to see all the presented data in full, we have assembled an archive file containing all the original images shown in the Figures, properly catalogued by panel, unambiguously named and scaled (ImageJ metadata). For SMLM images, we include reconstructions and the localization files (list of all fluorophores coordinates obtained from the SMLM acquisition sequence in Image ThunderSTORM plugin format) that have been used to generate them. For PREM views, we provide full resolution images of all panels. In the case of multi-channel and correlative images, we provide all single channel images as well as the corresponding overlay image. To also address **points 3.3 and 3.4** from Reviewer #3, we are providing all PREM images without color augmentation in order to be as close as possible to the original data. This archive will be made available as a data file deposited on a public repository (Figshare) and linked from the article on Nature Communications' website. The link to the archive file (1.8 GB zip file, private link at this time) is: <https://figshare.com/s/7cf4262c6d922add64ac>.

1.5 - *The manuscript is quite short, likely because it was transferred from a journal with rigorous space constraints? Anyway, it would profit from an expanded description and discussion of related work, especially the differences observed in contrast to the Jones et al (JCB 2014) study and possible implications regarding the fixation method and accessibility/visibility in the alternative approaches.*

We apologize for the brevity of our initial manuscript – indeed, this manuscript was transferred from another Journal. We have now taken advantage of the Nature Communications format and significantly expanded several points in our revised manuscript in addition to the new data. In particular, we have discussed how and why our results differ from the ones of Jones et al. (Jones et al., 2014), including an explicit reference to Zhong et al (Zhong et al., 2014). who showed that the live-cell extraction used in Jones et al. PREM protocol was disrupting the actin rings and MPS. This is now mentioned in the revised Discussion section, **p.6 lines 180-183**: “*Previous work using PREM to visualize the AIS organization did not detect regularly spaced actin filaments or other ultrastructural signs of the MPS. However, this work used live extraction by a detergent before fixation, an approach that was later shown to disrupt actin rings and the MPS.*”

Reviewer #2

This study examines periodic actin rings in axons of cultured neurons that have had their plasma membranes mechanically removed using both super-resolution light microscopy and platinum replica EM as well as correlative super-resolution light/platinum replica EM. The approach of mechanically detaching the cortical plasma membrane was introduced by John Heuser and allows for better preservation of internal cellular detail than current approaches of detergent extraction, where the plasma membrane collapses and can obscure high resolution imaging. An interesting new finding here is that the actin filaments are actually two

filaments that entwined or “braided”. I am not aware that this configuration of actin filaments has been reported before. I took the liberty of consulting with Tom Pollard who is highly informed regarding actin. He of course knew about actin filament bundles but not just two actin filaments. So, I believe that this is a “first”.

We would like to thank Reviewer #2 for this appreciation of our work, and for consulting Tom Pollard about actin braids. To the best of our knowledge, we indeed believe that actin rings as braids of two filaments are a novel cellular actin structure, as we mention in the revised manuscript **p.3, lines 75-76**: “*The organization of rings as braids of long, unbranched filaments is to our knowledge unique among cellular actin structures*”.

As the authors point out, such braided filaments would be more stable and resist depolymerization. Missing from the study is a mechanism for actin filament association. It would be of considerable interest if the authors could examine adducin-deficient axons, as adducin could be the “entwining” factor. However, this experiment, in my view, is not required for publication as a Nature Communication.

We agree with Reviewer #2 that the role of adducin in actin braid formation maintenance and association with the spectrins is a key point. As explained above in the answer to Reviewer #1, we were not successful so far in labeling adducin along unroofed proximal axons. We could also not set up experiments using genetic models such as adducin-KO mice in the time frame and scope of the revision of this manuscript. We are planning on developing ways to perturb molecular partners of the actin rings in future studies, and hope that this will bring additional insight in our follow-up work. Regarding the role of adducin, we have developed the discussion about its localization and role, **p. 6 lines 188-196**: “*This short filament model was inferred from the presence of adducin at actin rings, at least along the distal axon, because adducin was described as a capping protein that can bind to the barbed end of filaments. However, adducin is also able to associate to the side of filaments, enhancing the lateral binding of spectrin to actin. Although we were not able to directly localize adducin at the ultrastructural level, our results suggest that this lateral binding role is dominant to enhance actin rings interaction with spectrins. Further studies are needed to clarify the precise localization and role of adducin*”.

Overall, this is a carefully executed and well written paper. I have only a minor comment.

2.1. *It would be helpful to further support the two-filament model to include histograms of the widths of the “braided” filaments, and the individual filaments that diverge at “y” junctions, as well as microtubules.*

This point is important, and also made by Reviewer #3 (**point 3.1**). To address it, we carefully measured the thickness (using five measurement points along the length of the visible braid) for almost a hundred individual actin braids (see **Supplementary Table 3**), including those exhibiting an Y junction when splitting (**Fig. 2d-e**). Braids were 18.5 nm thick, splitting into two 10.2 nm filaments at Y junctions. This is perfectly compatible with braids being composed of two actin filaments splitting into one, taking into account the thickness of the ~2 nm platinum coating. As controls, we measure the thickness of isolated actin filaments within dendrites (9.9 nm) and the thickness of axonal microtubules (31.0 nm). This quantification is presented in **Fig. 2f**, with the summary statistics present in **Supplementary Table 3** and the individual values in the **Extended Data** file. We have updated the Results section text with this quantification on **p. 3 lines 71-75**: “*We confirmed that braids are made of two actin filaments by measuring the diameter of individual actin braids before and after splitting (Fig. 2f): braids are 18.2 ± 0.3 nm thick. When splitting, they form two 10.2 ± 0.3 nm filaments, similar in diameter to single actin filaments in dendrites (9.9 ± 0.2 nm), and to the reported value of 9-11 nm for a single actin filament rotary-replicated with ~2 nm of platinum.*”

We think this quantification strongly supports our claim that actin braids are made of two actin filaments, and would like to thank Reviewer #2 and #3 for suggesting it.

Reviewer #3

To date, a controversy in the field of cellular neurobiology is the structural organization of the cytoskeleton in axons. Specifically, a landmark 2013 paper used super-resolution imaging to reveal a system of dramatic periodic actin/spectrin rings that assembled in axons. The rings were proposed to mechanically stabilize the axon and organize channels and other receptors along the axon shaft. However, other work using more classic EM methods from Jones et al. was unable to see this periodic actin assembly and instead argued for a more dense, compact actin meshwork in axons. Currently, it is unclear how the two arrangements for actin obtained by these different nanoscopic imaging methods relate to the true structure of the cytoskeleton in living axons. In this paper by Vassilopoulos et al., this question is revisited with metal replica EM, super-resolution imaging, and correlative super-resolution light and EM. Through a powerful set of elegant experiments, the authors present a new actin structure in the axon that appears to be composed of submicron links of two twisted actin fibers that are rimmed by perpendicular septin fibers. The molecular identity of these elements are confirmed by immunogold EM in metal replicas. The authors go on to show that these structures can be disrupted by drugs. Finally, the authors argue with correlative methods that the stripes of actin observed in super-resolution fluorescence correspond to these actin fibers. In general, I think this paper is extremely interesting, with beautiful images, and thoughtful and careful analysis and experiments. It is a nice addition to the field of neuronal cell biology and will act as a catalyst for future work into the cytoskeleton and other structures found in neurons. I do, however, have several comments and concerns that should be addressed in a future version of the manuscript and would substantially improve the work. My specific comments follow.

We would like to thank Reviewer #3 for appreciating our work. We have tried our best to address the points raised below in the revised version of our manuscript.

3.1. *The paper's primary finding is that these unique actin filaments are stretched perpendicular to the axon and form relatively long "twisted" fibers that sometime splay into a "Y" shape. These fibers are arranged at an interval of around ~200 nm which nicely fits with the super-resolution spacing. However, because of a lack of fine-scale detail in the EM images, I cannot see the "twisting" or evidence that there are more than one actin fiber in these filaments. Finally, the splaying at the ends could be branched filaments that contact the primary filament; a common arrangement in the actin cortex. Why do the authors think these filaments are braided or twisted? Because this is such a unique structure for actin (the authors state "this organization of actin as braids of long, unbranched filaments is to our knowledge unique and contrasts with the currently assumed view that axonal actin rings made of short, capped and bundled actin filaments") the reader should have confidence in this new finding. From my eye, I cannot unmistakably see this structure. Are the filaments twice as wide as expected? Is there a regular expected braid twist seen along the filament that can be measured and isn't present in the other actin filaments in the same sample? This analysis would control for uneven metal coating and drying issues. I think a bit more is needed here to make for a strong argument for a totally new "twisted" fiber arrangement. This is such a key and novel part of the paper it would help to strengthen the evidence.*

We would like to thank the reviewer for raising this point that is indeed critical to our work. Let us address the several aspects raised in **point 3.1** and explain what leads us to propose that actin rings are made of two long, braided actin filaments.

Thickness of the actin braids. This point is important, and the suggestion of Reviewer #3 to measure actin braid thickness is in line with a similar concern from Reviewer #2 (**point 2.1**). We reproduce here part of our response to this point above: [to address it, we carefully measured the thickness (using five measurement points along the length of the visible braid) for almost a hundred individual actin braids (see **Supplementary Table 3**), including those exhibiting an Y junction when splitting (**Fig. 2d-e**). Braids were 18.5 nm thick, splitting into two 10.2 nm filaments at Y junctions. This is perfectly compatible with braids being composed of two actin filaments splitting into one, taking into account the thickness of the ~2 nm platinum coating. As controls, we measure the thickness of isolated actin filaments within dendrites (9.9 nm) and the thickness of axonal microtubules (31.0 nm). This quantification is presented in **Fig. 2f**, with the summary statistics present in **Supplementary Table 3** and the individual values in the **Extended Data** file. We have updated the Results section text with this quantification on **p. 3 lines 71-75**: “*We confirmed that braids are made of two actin filaments by measuring the diameter of individual actin braids before and after splitting (Fig. 2f): braids are 18.2 ± 0.3 nm thick. When splitting, they form two 10.2 ± 0.3 nm filaments, similar in diameter to single actin filaments in dendrites (9.9 ± 0.2 nm), and to the reported value of 9-11 nm for a single actin filament rotary-replicated with ~2 nm of platinum.*” We think this quantification strongly supports our claim that actin braids are made of two actin filaments, and would like to thank Reviewer #2 and #3 for suggesting it.]

Alternative interpretation as classical cortical actin branching organization. As pointed out by Reviewer #3, an alternative interpretation to the actin ring splitting (**Fig. 2d-e**) could be side branching of an actin filament, such as those mediated by Arp2/3. In this case, one would expect the thickness along a “splitting” ring to be constant before and after the split, with all branches having the ~10 nm thickness of a platinum-replicated actin filament. This is what is observed on PREM views of Arp2/3-branched actin networks, where the mother filament and the branch have the same thickness (Ferrari et al., 2019; Franck et al., 2019). In the case of actin rings (see above), the braid is twice the size of each split branch, and each split branch has the same thickness as single actin filaments (as measured in dendrites). We think this rules out side-branching and demonstrate that rings are made of two filaments.

Evidence for the braiding of the two filaments. From the numerous examples of actin rings that we have observed by PREM, we are convinced that the two actin filaments that compose them are intertwined together. We have provided several examples of such braiding in the gallery of **Fig. 2d-e**. We have reproduced several of these at high magnification in the **Reviewer Figure 5** below, with an additional example. What strongly suggests that the two filaments are braided together is the regular variation in thickness along the ring – it is the most likely arrangement of two filaments that could generate such a pattern. We have tried

to color the braids on the four examples given. It is nevertheless difficult to precisely quantify the step size of the braid (Reviewer Figure 5).

Reviewer Figure 5. Examples of actin rings with braiding colored

Examples 1-3 are reproduced from Fig. 2d. Perceived position of each filaments within the rings are shown on the right of each example in blue and red. Additional example 4 shows the succession of minima and maxima in thickness (brown and orange arrowheads respectively). Scale bars 100 nm.

3.2. The correlative work, while impressive, is missing most of the unique actin structures. Thus, I am not sure how much this figure is adding to the story. It appears that the key actin filaments were lost or destroyed during the fluorescence imaging or maybe didn't survive the shipping for EM? Regardless, the remaining material left after the fluorescence imaging is hardly visible in EM. I can only see a few matching filaments in these images. Some of the images have no filaments that correspond to the fluorescence signal. I am not sure how this could be quantitatively analyzed to support the author's argument but it might be nice to provide some statistical measurement of how often filaments corresponded to the fluorescence or alternatively how often EM filament had fluorescence.

This is also a very important point and we are thankful to Reviewer #3 for bringing it up. First, we would like to respectfully emphasize that we think the correlative experiments represent crucial data to unambiguously equate the actin rings seen by STORM and the actin braids seen in PREM, as mentioned by Reviewer #1: “The data are presented in an excellent fashion and allow access to both EM data and EM-SMLM data in parallel, a very well-documented and excellently executed verification of the experimental system and convincing identification of the respective components.” In addition, these experiments allow us to precisely answer why in unroofed neurons, the rings seen by SMLM and braids seen by PREM show similar patterns with some key differences: the SMLM images usually show a large number of long, regularly spaced rings, whereas PREM only shows small portions of them (Fig. 5a and 5d vs Fig. 5b and 5d). When precisely comparing actin filaments between the SMLM and PREM images, there are several reasons why some discrepancy can exist:

A filament is present on the PREM images and absent from the SMLM image. Firstly, the SMLM reconstructed image size is often smaller than the PREM image used for overlay, so there are some panels with no fluorescence signal on their borders (always outside of the axon considered). Secondly, we used a fast staining protocol to perform unroofing, fixation, labeling and STORM imaging within one day, in order to ship coverslips at the end of the day and prepare platinum replica on the next day, which best preserved the sample ultrastructure. For this reason, the actin labeling by phalloidin is not as dense as in a regular SMLM experiment where we label using phalloidin at 4°C overnight, and some filaments seen on PREM images

are only appearing on the SMLM image as a handful of localizations (dots). This can be seen for some filaments in **Fig. 5**.

A filament is present on the SMLM image and absent from the PREM image. Firstly, when labeling other components than actin (such as β 4-spectrin in **Fig. 6** and **Supplementary Fig. 5**), the high density of staining necessary to obtain a good SMLM image required us to use a high concentration of antibodies, which results in the spectrin mesh and actin braids often being obscured by a large quantity of bound IgGs. This can be most clearly seen on **Supplementary Fig. 5d-f**, which was the original correlative β 4-spectrin data shown in the initial version of this manuscript, where the mesh has a granular appearance as opposed to the fibrous pattern seen in unlabeled samples (for example **Fig. 2g**). We did a new series of experiments for correlative β 4-spectrin SMLM/PREM, using a lower concentration of antibodies, and obtained a better preservation of the mesh and rings appearance. A representative example from these new experiments is now included in **Fig. 6a-c** and **Supplementary Fig. 5a-c**.

Secondly, as mentioned above, SMLM images of phalloidin-stained actin usually show a large number of long, regularly spaced rings, whereas PREM only shows small portions of them (**Fig. 5a and 5d** vs **Fig. 5b and 5d**). By analyzing the ultrastructural views in depth, we notice that many actin filaments that are continuous on the SMLM image are hidden between the plasma membrane and the spectrin mesh, or behind microtubules on the PREM view. We think this is the main reason why many filaments are not visible or only partially on the PREM view compared to the SMLM image. We are now explaining this in the revised text of the Results section, **p. 5 lines 140-141 and 145-155**: “*The braids seen by PREM are not as long and continuous along the axon as the actin rings visible in SMLM (compare Fig. 1b and 1j-k). [...] Phalloidin staining of actin rings on SMLM images (Fig. 5a and 5d) frequently corresponded with the braids visible on PREM views (Fig. 5b and 5e). A few actin filaments on PREM images were only faintly stained by phalloidin, likely due to the fast labeling protocol used for our correlative approach. SMLM often showed single rings across the whole width of the axon, whereas they only appeared intermittently on the PREM views (Fig. 5c and 5f, Supplementary Movie 5). This may result from ultrastructural damage during SMLM imaging or sample processing for subsequent PREM. However, close examination of PREM views - that only visualize the surface of the replicated sample - suggests that phalloidin-stained actin rings are often embedded between the plasma membrane and the spectrin mesh, and thus partially hidden from PREM view (Fig. 5c and 5f). Phalloidin staining also revealed that rings were continuous under microtubules, although they are similarly hidden from PREM view (Fig. 5c and 5f).*”

As suggested by Reviewer #3, we cannot exclude that a few filaments have been damaged during SMLM imaging and sample shipping before PREM. We thus added this point in the Results section on **p. 5, lines 150-151**: “*This may result from ultrastructural damage during SMLM imaging or sample processing for subsequent PREM*”. We could not devise a quantification procedure to obtain hard numbers on the correlation between SMLM and PREM, as this is a difficult task. However, we have performed an additional series of actin labeling/correlative PREM experiments to strengthen this data, and added a second representative example from these experiments, **Fig. 5d-f**.

As it stands now, I don't feel like this figure is strongly supporting the main arguments of the paper over and above what is shown in the super-resolution alone, EM alone, and immunogold EM images alone. Technically, this figure is an amazing achievement but the data just don't look that convincing.

We appreciate that Reviewer #3 finds our CLEM analysis an amazing achievement, a first in neurons to the best of our knowledge. We do hear the remark concerning the usefulness of such an experiment for our

demonstration. As mentioned above, some confusion as to the quality of the correlative imaging stems from the fact that a significant proportion of filaments are buried underneath the spectrin mesh. This is a result in itself and in our view, justifies the correlative approach we used to characterize the axonal actin rings. It explains why even when all the rings are labeled by phalloidin as seen by SMLM, we cannot always visualize the underlying actin structures by PREM. Without using CLEM to visualize these structures, we could not have revealed this specific characteristic of axonal actin rings.

3.3. *It would be nice to provide images of the immuno-gold that did not have color augmentation.*

3.4. *It would be nice to always have corresponding images in the EM figures that do not have color augmentation so that the reader can evaluate the structures without the input of the authors.*

As mentioned when replying to Reviewer #1 about making high-resolution images available (**point 1.4**), we have now prepared a data file containing all PREM images at native resolution with no pseudo-coloring. After consulting with the Editors about the best way to make this data available, this archive will be deposited on a public repository (Figshare) and linked from Nature Communications' website. The link to the archive file (1.8 GB zip file, private link at this time) is <https://figshare.com/s/7cf4262c6d922add64ac>.

3.5. *How many cells were imaged for EM? This information is not present as far as I can tell in the document. How often were these actin fibers seen? Was it common or rare? This is important information to put in the main text or figure legend for the reader. I would like to see an “n” for number of cells, number of EM preps, and number of structures observed. This information was given for the fluorescent data but not EM or CLEM.*

We agree with Rev #3 and have now included all the information concerning number of axons imaged for each condition in Table 2 of the main manuscript. However, it is important to state that actin braids were present in every unroofed axon imaged. We have modified the text of the Results section to clearly state this, **p.2 lines 55-56**: *“These structures, present in all axons observed by PREM, resembled braids made of actin filaments”*.

References

Abouelezz A, Stefen H, Segerstråle M, Micinski D, Minkeviciene R, Hardeman EC, Gunning PW, Hoogenraad C, Taira T, Fath T, Hotulainen P (2019) Tropomyosin Tpm3.1 is required to maintain the structure and function of the axon initial segment. bioRxiv.

Berger SL, Leo-Macias A, Yuen S, Khatri L, Pfennig S, Zhang Y, Agullo-Pascual E, Caillol G, Zhu M-S, Rothenberg E, Melendez-Vasquez CV, Delmar M, Leterrier C, Salzer JL (2018) Localized Myosin II Activity Regulates Assembly and Plasticity of the Axon Initial Segment. *Neuron* 97:555–570.e556.

Ferrari R, Martin G, Tagit O, Guichard A, Cambi A, Voituriez R, Vassilopoulos S, Chavrier P (2019) MT1-MMP directs force-producing proteolytic contacts that drive tumor cell invasion. *Nat Commun* 10:4886.

Franck A, Lainé J, Moulay G, Lemerle E, Trichet M, Gentil C, Benkhelifa Ziyat S, Lacene E, Bui MT, Brochier G, Guicheney P, Romero N, Bitoun M, Vassilopoulos S (2019) Clathrin plaques and associated actin anchor intermediate filaments in skeletal muscle. *Mol Biol Cell* 30:579–590.

Fréal A, Fassier C, Le Bras B, Bullier E, De Gois S, Hazan J, Hoogenraad CC, Couraud F (2016) Cooperative Interactions between 480 kDa Ankyrin-G and EB Proteins Assemble the Axon Initial Segment. *J Neurosci* 36:4421–4433.

Fréal A, Rai D, Tas RP, Pan X, Katrukha EA, van de Willige D, Stucchi R, Aher A, Yang C, Altelaar AFM, Vocking K, Post JA, Harterink M, Kapitein LC, Akhmanova A, Hoogenraad CC (2019) Feedback-Driven Assembly of the Axon Initial Segment. *Neuron*.

Galiano MR, Jha S, Ho TS-Y, Zhang C, Ogawa Y, Chang K-J, Stankewich MC, Mohler PJ, Rasband MN (2012) A distal axonal cytoskeleton forms an intra-axonal boundary that controls axon initial segment assembly. *Cell* 149:1125–1139.

Jenkins PM, Kim N, Jones SL, Tseng W-C, Svitkina TM, Yin HH, Bennett V (2015) Giant ankyrin-G: a critical innovation in vertebrate evolution of fast and integrated neuronal signaling. *Proc Natl Acad Sci USA* 112:957–964.

Jones SL, Korobova F, Svitkina T (2014) Axon initial segment cytoskeleton comprises a multiprotein submembranous coat containing sparse actin filaments. *J Cell Biol* 205:67–81.

Leterrier C, Potier J, Caillol G, Debarnot C, Rueda-Boroni F, Dargent B (2015) Nanoscale Architecture of the Axon Initial Segment Reveals an Organized and Robust Scaffold. *Cell Rep* 13:2781–2793.

Leterrier C, Vacher H, Fache M-P, Angles d'Ortoli S, Castets F, Autillo-Touati A, Dargent B (2011) End-binding proteins EB3 and EB1 link microtubules to ankyrin G in the axon initial segment. *Proc Natl Acad Sci USA* 108:8826–8831.

Wang T, Li W, Martin S, Papadopoulos A, Jiang A, Shamsollahi G, Amor R, Lanoue V, Padmanabhan P, Meunier FA (2018) Radial contractility of Actomyosin-II rings facilitates cargo trafficking and maintains axonal structural stability following cargo-induced transient axonal expansion. *bioRxiv*.

Xu K, Zhong G, Zhuang X (2013) Actin, spectrin, and associated proteins form a periodic cytoskeletal structure in axons. *Science* 339:452–456.

Zhong G, He J, Zhou R, Lorenzo D, Babcock HP, Bennett V, Zhuang X (2014) Developmental mechanism of the periodic membrane skeleton in axons. *Elife* 3:194.

Reviewers' Comments:

Reviewer #1:

Remarks to the Author:

The authors did a significant amount of additional work that further enlightens our view of the structural organization of the axonal initial segment. The authors have addressed my main concerns and I have but minor further suggestions that do not prohibit publication.

The authors show striking data on alignment of multiple AnkG tails along microtubules, which wonderfully supports their earlier work on a connection of AnkG via EB1/3 to the microtubule cytoskeleton. However EB1/3 is not mentioned at all in the manuscript. The reviewer believes it would be useful to reference the nature of the link of AnkG to microtubules in the discussion.

Reviewer #2:

Remarks to the Author:

The authors have addressed my concerns and also provided additional data on phosphorus-myosin light chain and AnkG-480. In view of the connection between ankG-480 and microtubules they may wish to cite reports of loss of microtubule bundling at the AIS in AnkG-deficient neurons by Sobotznic et al 2009 and Jenkins et al 2015.

Reviewer #3:

Remarks to the Author:

The authors have addressed all the concerns from my first review in this new revision. It is now a lovely paper.

Response to reviewers for “Ultrastructure of the axonal periodic scaffold reveals a braid-like organization of actin rings”, manuscript NCOMMS-19-20620-A

S. Vassilopoulos, S. Gibaud, A. Jimenez, G. Caillol, C. Leterrier

We would like to thank the three Reviewers for the final assessment of our manuscript and helpful remarks and suggestions. Based on their feedback, we have revised our manuscript to address the points raised.

Reviewer #1

The authors did a significant amount of additional work that further enlightens our view of the structural organization of the axonal initial segment. The authors have addressed my main concerns and I have but minor further suggestions that do not prohibit publication.

The authors show striking data on alignment of multiple AnkG tails along microtubules, which wonderfully supports their earlier work on a connection of AnkG via EB1/3 to the microtubule cytoskeleton. However EB1/3 is not mentioned at all in the manuscript. The reviewer believes it would be useful to reference the nature of the link of AnkG to microtubules in the discussion.

We added a mention to EB1 and EB3 being the link between ankyrin G and microtubules when discussing the results of the immunogold and correlative experiments in the Results section.

Reviewer #2

The authors have addressed my concerns and also provided additional data on phosphorus-myosin light chain and AnkG-480. In view of the connection between ankG-480 and microtubules they may wish to cite reports of loss of microtubule bundling at the AIS in AnkG-deficient neurons by Sobotznic et al 2009 and Jenkins et al 2015.

We have added the citations suggested when presenting the role of ankyrin G at the AIS in the Results section.

Reviewer #3

The authors have addressed all the concerns from my first review in this new revision. It is now a lovely paper.

We would like to thank Reviewer #3 for appreciating our work.